# A dynamic calcium-force relationship model for sag behavior in fast skeletal muscle

**Hojeong Kim**[1,2]*, **Charles J. Heckman**[3,4,5]

**1** Division of Biotechnology, Institute of Convergence Research, DGIST, Daegu, Republic of Korea,
**2** Department of Interdisciplinary Studies, DGIST, Daegu, Republic of Korea, **3** Department of Neuroscience, Feinberg School of Medicine, Northwestern University, Chicago, Illinois, United States of America, **4** Physical Therapy and Human Movement Sciences, Feinberg School of Medicine, Northwestern University, Chicago, Illinois, United States of America, **5** Physical Medicine and Rehabilitation, Feinberg School of Medicine, Northwestern University, Chicago, Illinois, United States of America

* Hojeong.kim03@gmail.com

**Data Availability Statement:** All relevant data are within the manuscript and its Supporting Information files. The codes created for this study are available in Supporting Information and the

## Abstract

In vitro studies using isolated or skinned muscle fibers suggest that the sigmoidal relationship between the intracellular calcium concentration and force production may depend upon muscle type and activity. The goal of this study was to investigate whether and how the calcium-force relationship changes during force production under physiological conditions of muscle excitation and length in fast skeletal muscles. A computational framework was developed to identify the dynamic variation in the calcium-force relationship during force generation over a full physiological range of stimulation frequencies and muscle lengths in cat gastrocnemius muscles. In contrast to the situation in slow muscles such as the soleus, the calcium concentration for the half-maximal force needed to drift rightward to reproduce the progressive force decline, or sag behavior, observed during unfused isometric contractions at the intermediate length under low-frequency stimulation (i.e., 20 Hz). The slope at the calcium concentration for the half-maximal force was required to drift upward for force enhancement during unfused isometric contractions at the intermediate length under high-frequency stimulation (i.e., 40 Hz). The slope variation in the calcium–force relationship played a crucial role in shaping sag behavior across different muscle lengths. The muscle model with dynamic variations in the calcium-force relationship also accounted for the length-force and velocity-force properties measured under full excitation. These results imply that the calcium sensitivity and cooperativity of force-inducing crossbridge formation between actin and myosin filaments may be operationally altered in accordance with the mode of neural excitation and muscle movement in intact fast muscles.

## Author summary

The intracellular calcium concentration is a crucial factor determining the magnitude of active force in skeletal muscle. The relationship between muscle force and intracellular calcium concentration has been characterized by observing the force responses of isolated or skinned muscle fibers at different calcium concentration levels under controlled in

public repository of ModelDB (http://modeldb.yale.edu/267738).

**Funding:** HK received funding from the National Research Foundation of Korea (NRF, https://www.nrf.re.kr) (2021060027) and the Ministry of Science and ICT (MSIT, https://www.msit.go.kr) (DGIST R&D Program 21-BT-06) and CJH received funding from the National Institutes of Health (NIH, https://www.nih.gov) (NS071951 and NS062200). The funders had no role in study design, data collection and analysis, decision to publish, or preparation of the manuscript.

**Competing interests:** The authors have declared that no competing interests exist.

vitro conditions. However, it remains unclear whether and how the calcium-force relationship changes under physiological conditions, including neural excitation and muscle movement. Because of current experimental limitations, we have developed a computational framework that allows us to identify the calcium-force relationship for force generation over a full physiological range of stimulation frequencies and muscle lengths in cat medial gastrocnemius muscles. The results indicate that, in contrast to the situation in slow muscles such as the soleus, dynamic variation in the calcium-force relationship is required to reproduce the complex force profiles observed in fast skeletal muscles under physiological conditions. This study may not only provide insights into the dynamic properties of calcium-mediated crossbridge formation between thin and thick filaments for force generation but also offer a computational framework for realistic muscle modeling.

## Introduction

Skeletal muscles are specialized tissues that can contract to generate force in response to neural signals from spinal motoneurons. It is commonly thought that contractile forces are induced from crossbridge formation between actin and myosin filaments in the sarcomere [1–4], which is activated by calcium release from the sarcoplasmic reticulum according to neural excitation [5]. Thus, the transfer function of force inducing crossbridge formation across different excitation levels has been characterized by measuring steady-state force responses at various constant calcium concentration levels [6].

Isolated or skinned muscle fiber experiments on the calcium sensitivity of muscle force have shown larger force production at higher calcium concentrations compared to the lower constant calcium concentration levels at steady state [7]. This sigmoid steady-state calcium-force relationship implies the cooperativity and saturation of crossbridge formation as the level of muscle excitation increases [8–10]. The steady-state calcium-force relationship has also been shown to be specific to the type of muscle fiber (slow-twitch versus fast-twitch in [11]) in mammals [12–16] and humans [17]. Recent in vitro studies have further demonstrated that the shape of the calcium-force relationship can be considerably altered by inorganic phosphate molecules known to accumulate during force production [18] or variation in muscle length [7], presumably owing to changes in actin-myosin interaction [19]. However, it remains elusive whether and how the calcium-force relationship varies in intact muscle fibers under physiological conditions of muscle excitation and length.

Computational studies have suggested that the calcium-force relationship might differ depending on the type of muscle fibers (slow-twitch versus fast-twitch) under physiological conditions. Slow-twitch muscles have been shown to produce a force whose peak is maintained during isometric contractions at any level of stimulation frequency [20]. These isometric forces could be reproduced for a full physiological range of stimulation frequencies with no changes in the calcium-force relationship in computational models [21]. In contrast, fast-twitch muscles have been shown to produce a force that declines rapidly within half a second after its peak during isometric contractions under unfused stimulation conditions [22]. The impaired crossbridge formation by the inhibitory effects of inorganic phosphate accumulation has been computationally and experimentally demonstrated as a potential mechanism for this rapid force decline (or sag) during continuous submaximal contractions [23]. On the other hand, the decline in free calcium ion concentration in the sarcoplasm by the reduction in calcium release from the sarcoplasmic reticulum [24] and the accelerated uptake of calcium into the sarcoplasmic reticulum [25] have been proposed to be involved in the final phase of muscle

fatigue, showing accelerated force reduction several seconds after the onset of continuous maximal contraction [24,26].

Here, we focus on the dynamic variation in the calcium-force relationship during the rapid force decline (or sag) in fast skeletal muscles before the final phase of muscle fatigue. It is hypothesized that both the midpoint and the slope of the sigmoid calcium-force relationship dynamically change for force production under physiological conditions. Because of current experimental limitations, a computational approach is developed to test our hypothesis. The comparison between experimental and simulation results shows that dynamic variation in the calcium-force relationship is required to replicate the force responses of the cat medial gastrocnemius over the full physiological range of stimulation frequencies and muscle lengths. This study provides insights into the dynamic properties of calcium-sensitive crossbridge formation under physiological conditions and a computational framework for realistic modeling of complex muscle behaviors.

## Materials and methods

### Ethics statement

All procedures for collecting data from cat medial gastrocnemius (MG) were approved by the Animal Care Committee at Northwestern University.

### In situ experiments

The properties of average active force production by single sarcomeres were characterized by measuring the active force of the whole muscle in decerebrated cats [3]. Details of the surgical preparations and experimental protocols have been previously reported [27]. Briefly, adult cats (CAT14 for model development and CAT12 for prediction confirmation) were anesthetized, the medial gastrocnemius was exposed, and the distal tendon was attached to a computer-controlled puller. All distal hindlimb nerves except the medial gastrocnemius nerve were disconnected, only ipsilateral dorsal roots from the fourth lumbar spinal nerve (L4) to the second sacral spinal nerve (S2) were transected to eliminate sensory feedback from the medial gastrocnemius, and decerebration was performed at the midbrain. Hindlimb and core temperatures were maintained within physiological limits. Electrical impulses with a width of 0.1 ms were applied to control muscle force either using stainless steel wires in the proximal and distal portions of the muscle belly or via hook electrodes on the ventral roots. A supra axial current (50–100% above that required to elicit full recruitment) was used to produce repeatable and consistent forces during all trials. The length of the muscle-tendon unit was controlled by a position servo system and measured by a linear variable differential transformer attached to the puller shaft. Muscle forces were measured by a strain gauge-based transducer in series with the shaft. Raw data for muscle force and length were smoothed using the function ("smooth" with options of "0.01" for span and "loess" for method) built in MATLAB software (version R2012a, MathWorks, Natick, MA). To isolate the active force response, the force data measured without the stimulation (i.e., passive response) were subtracted from those measured with the stimulation (i.e., total response) under the same condition of muscle-tendon length. The passive and active trials were separated by a 30 s rest period, and all active trials were also separated by at least one minute to minimize fatigue.

### Input protocols

The calcium-force relationship was dynamically evaluated during twitch, unfused and fused tetani produced with constant stimulation frequencies (i.e., 1, 10, 20, 40 and 100 Hz) at the

intermediate muscle-tendon length ($X_{m,0.5}$, 5 mm for CAT14 and 3.64 mm for CAT12) between the physiologically minimal ($X_{m,0}$, 0 mm for CAT14 and -0.55 mm for CAT12) and maximal ($X_{m,1}$, 10 mm for CAT14 and 7.9 mm for CAT12) lengths that have been observed during locomotor-like movements in cats [28–30]. The representative types of sag behavior were reproduced under constant stimulation frequencies of 20 and 30 Hz at $X_{m,0.5}$ for both CAT14 and CAT12. The length and velocity dependence of the dynamics of the calcium-force relationship was investigated during shortening and lengthening at constant velocities and step lengthening over a very short period of time under full excitation (100 Hz) for both CAT14 and CAT12.

## Modular modeling framework

In this study, we used a modular modeling approach that allowed us to divide the complex process of force generation into subprocesses that can be separately modeled and validated by relevant experimental data. A modular model of the muscle-tendon complex experimentally validated for the adult cat soleus (predominantly slow twitch fibers [20]) in in situ conditions was adapted to replicate the force production of the adult cat medial gastrocnemius (predominantly fast twitch fibers [31]). The derivation and description of model equations for the model of slow skeletal muscles has been fully presented in a previous study [21]. We employed Hill-type mechanics comprising a contractile and an elastic element in series [32] because it is simple and effective for modular modeling of the essential processes that intact muscle undergoes for contractile force production under physiological conditions (Fig 1). The first module (i.e., Module 1) transforms the electrical signals coming from either spinal motoneurons or electrical stimulation into the concentration of free calcium (Ca) within the sarcoplasm (SP). The concentration of sarcoplasmic calcium was determined by multiple factors, including the reaction of calcium ($Ca_{SR}$) and calsequestrin (CS) in the sarcoplasmic reticulum (SR), calcium release (R) and uptake (U) through the membrane of the SR, calcium buffering by free proteins (B), and calcium-troponin binding (CaT) modulated by the muscle-tendon length ($X_m$) and muscle activation level ($\tilde{A}$) in terms of the constant CaT. The system equations for Module1 are as follows:

$$\frac{d[Ca_{SR}]}{dt} = -K1 \cdot CS_0 \cdot [Ca_{SR}] + (K1 \cdot [Ca_{SR}] + K2) \cdot [Ca_{SR}CS] - R + U \tag{1}$$

$$\frac{d[Ca_{SR}CS]}{dt} = K1 \cdot CS_0 \cdot [Ca_{SR}] - (K1 \cdot [Ca_{SR}] + K2) \cdot [Ca_{SR}CS] \tag{2}$$

$$\frac{d[Ca]}{dt} = -(K3 \cdot B_0 + K5 \cdot T_0) \cdot [Ca] + (K3 \cdot [Ca] + K4) \cdot [CaB] + (K5 \cdot [Ca] + K6) \cdot [CaT]$$
$$+ R - U \tag{3}$$

$$\frac{d[CaB]}{dt} = K3 \cdot B_0 \cdot [Ca] - (K3 \cdot [Ca] + K4) \cdot [CaB] \tag{4}$$

$$\frac{d[CaT]}{dt} = K5 \cdot T_0 \cdot [Ca] - (K5 \cdot [Ca] + K6) \cdot [CaT] \tag{5}$$

where $CS_0$, $B_0$ and $T_0$ indicate the concentration of total calsequestrin in SR, total free calcium-buffering proteins, and total troponin in SP, respectively, and *K1-K6* are the rate constants for chemical reactions between $Ca_{SR}$, $Ca_{SR}$CS, Ca, B, T, CaB and CaT.

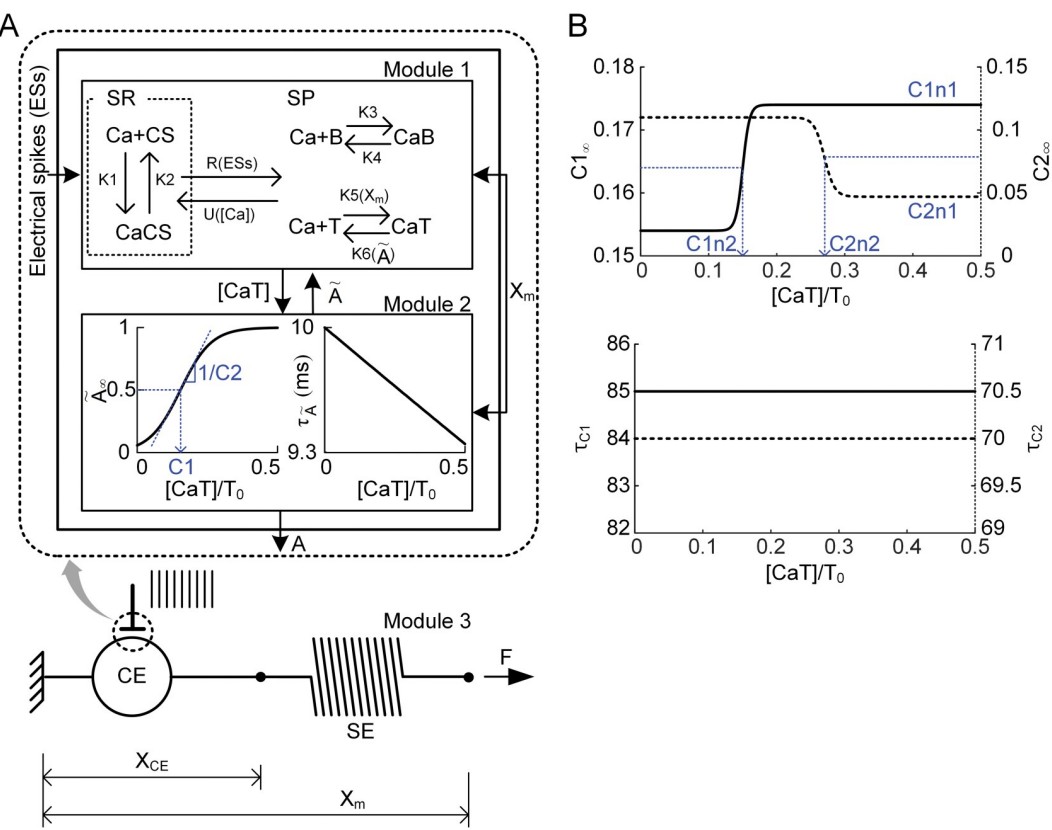

**Fig 1. Schematic diagram of the action potential-driven muscle-tendon model. A.** Muscle activation dynamics (A) in response to action potentials (ESs) from spinal motoneurons (dotted line box) and muscle mechanics (F) induced by contractile elements (CEs) and serial elastic elements (SEs). $X_{CE}$ and $X_m$ show the length of the contractile element and entire muscle-tendon unit. C1 and 1/C2 indicate the constant concentration of calcium-binding troponin relative to the total troponin concentration (CaT/$T_0$) for the half-maximal level of steady-state muscle activation ($\tilde{A}_\infty$) and the slope of the CaT/$T_0$–$\tilde{A}_\infty$ curve at C1. **B.** Relationship of steady-state C1 ($C1_\infty$) and C2 ($C2_\infty$) with the concentration of CaT/$T_0$ (upper panel) and time constants ($\tau_{C1}$ and $\tau_{C2}$) for the C1 and C2 dynamics (bottom panel). *C1n1* and *C2n1* limit the saturation level, and *C1n2* and *C2n2* indicate a constant concentration of CaT/$T_0$ for the midpoint between the minimum and maximum in the $C1_\infty$ and $C2_\infty$ curves.

The action of the calcium pump in the SR was mathematically modeled as R and U in the absence of an inhibitor for pump action [33],

$$R = [Ca_{SR}] \cdot R_{max} \cdot \sum_{i=1}^{n} \left( 1 - exp\left( -\frac{t - t_i}{\tau_1} \right) \right) \cdot exp\left( -\frac{t - t_i}{\tau_2} \right)$$

$$U = U_{max} \cdot \left( \frac{[Ca]^2 \cdot K^2}{1 + [Ca] \cdot K + [Ca]^2 \cdot K^2} \right)^2$$

where $R_{max}$, $\tau_1$, $\tau_2$, $U_{max}$, and $K$ indicate the maximum calcium release rate from the SR, the time constant of calcium release increase, the time constant of calcium release decrease, the maximal calcium uptake rate, and the association constant for calcium binding to pump, respectively.

To reflect the length-dependent changes in the calcium-force relationship [34], the default value for the forward rate constant (i.e., $K5_i$) of the calcium-troponin reaction was made as a

function of $X_m$ to match twitch responses at different $X_m$ values, as suggested in a previous study [35],

$$K5 = K5_i \cdot \phi(X_m), \begin{cases} \phi(X_m) = \phi_1 \cdot X_m + \phi_2, for\ X_m < X_{m,0.5} \\ \phi(X_m) = \phi_3 \cdot X_m + \phi_4, for\ X_m \geq X_{m,0.5} \end{cases}$$

In addition, the affinity increase of troponin for calcium by crossbridge attachments [36] was reflected by varying the default value for the backward rate constant (i.e., $K6_i$) inversely to $\tilde{A}$ in a similar manner as reported in a previous study [33],

$$K6 = \frac{K6_i}{1 + 5 \cdot \tilde{A}}$$

The second module (i.e., Module 2) transforms the concentration of CaT relative to the total troponin concentration ($T_0$) into the level of muscle activation (A), representing a fraction of actomyosin crossbriges available for force generation. A was determined through exponentiation of $\tilde{A}$ with a single exponent ($\alpha$) to reflect the degradation of muscle activation for the fluctuation in CaT under physiological conditions. The system equations for Module 2 are as follows:

$$A(t) = (\tilde{A})^\alpha \tag{6}$$

The dynamics of $\tilde{A}$ were mathematically represented,

$$\frac{d\tilde{A}}{dt} = \frac{\tilde{A}_\infty - \tilde{A}}{\tau_{\tilde{A}}} \text{ where } \tilde{A}_\infty = 0.5 \cdot \left(1 + tanh\frac{[CaT]/T_0 - C1}{C2}\right), \tau_{\tilde{A}}$$
$$= C3 \cdot \left(cosh\frac{[CaT]/T_0 - C4}{2 \cdot C5}\right)^{-1}$$

The third module (i.e., Module 3) transforms the muscle activation level (A) resulting from crossbridge formation into the force (F) based on Hill-type mechanics. F was determined via multiplication of the stiffness ($K_{SE}$) of serial elastic elements (SE), including extracellular (i.e., tendons and aponeuroses) and intracellular (i.e., titin and myofilaments) components, with the variation in SE length reflecting the length- and velocity-tension properties measured under full excitation. The SE length variation was calculated by subtracting a change in the contractile element ($\Delta X_{CE}$) from a change in the whole muscle-tendon unit ($\Delta X_m$) length. The length-tension relationship was proportionally adjusted to the level of muscle activation (A) resulting from neural excitation or electrical stimulation [37]. The tension axis of the velocity-tension curve was scaled by the length-tension relationship to reflect the overlap of sliding filaments determining the number of force-generating crossbridges [38]. The system equations for Module 3 are as follows:

$$F = P_{0.5} \cdot K_{SE} \cdot (\Delta X_m - \Delta X_{CE}) \tag{7}$$

where $P_{0.5}$ is the peak force at the intermediate length between the physiologically minimal and maximal length under full excitation in the isometric condition and $K_{SE}$ is the stiffness of the serial element normalized by $P_{0.5}$.

The $\Delta X_{CE}$ was calculated using the modified Hill-Mashma equations along with the length-tension relationship ($g(X_m)$),

$$\frac{dX_{CE}}{dt} = \frac{-b_0 \cdot (P_{0.5} \cdot g(X_m) \cdot A(t) - F)}{F + a_0 \cdot g(X_m) \cdot A(t)}, for\ F \leq P_{0.5} \cdot g(X_m) \cdot A(t)$$

$$\frac{dX_{CE}}{dt} = \frac{-d_0 \cdot (P_{0.5} \cdot g(X_m) \cdot A(t) - F)}{2 \cdot P_{0.5} \cdot g(X_m) \cdot A(t) - F + c_0 \cdot g(X_m) \cdot A(t)}, for\ F > P_{0.5} \cdot g(X_m) \cdot A(t)$$

$$g(X_m) = g_1 \cdot X_m^2 + g_2 \cdot X_m + g_3$$

Given the four data points (($V_{S,1}$, $T_{S,1}$), ($V_{S,2}$, $T_{S,2}$), ($V_{L,1}$, $T_{L,1}$), ($V_{L,2}$, $T_{L,2}$)) on the velocity-tension (V-T) curve, the Hill-Mashma coefficients ($a_0$, $b_0$, $c_0$ and $d_0$) can be analytically determined by the following inverse equations:

$$a_0 = \frac{V_{S,1} \cdot T_{S,1} \cdot (P_{0.5} - T_{S,2}) - V_{S,2} \cdot T_{S,2} \cdot (P_{0.5} - T_{S,1})}{V_{S,2} \cdot (P_{0.5} - T_{S,1}) - V_{S,1} \cdot (P_{0.5} - T_{S,2})}$$

$$b_0 = \frac{V_{S,2} \cdot V_{S,1} \cdot (T_{S,1} - T_{S,2})}{V_{S,1} \cdot (P_{0.5} - T_{S,2}) - V_{S,2} \cdot (P_{0.5} - T_{S,1})}$$

$$c_0 = \frac{(2 \cdot V_{L,2} \cdot P_{0.5} - V_{L,2} \cdot T_{L,2}) \cdot (P_{0.5} - T_{L,1}) + (V_{L,1} \cdot T_{L,1} - 2 \cdot V_{L,1} \cdot P_{0.5}) \cdot (P_{0.5} - T_{L,2})}{V_{L,1} \cdot \{P_{0.5} - T_{L,2} - V_{L,2} \cdot (P_{0.5} - T_{L,1})\}}$$

$$d_0 = \frac{V_{L,1} \cdot V_{L,2} \cdot (T_{L,1} - T_{L,2})}{V_{L,2} \cdot (P_{0.5} - T_{L,1}) - V_{L,1} \cdot (P_{0.5} - T_{L,2})}$$

where $V_{S,1}$ and $V_{S,2}$ are the minimum and maximum shortening velocities and $V_{L,1}$ and $V_{L,2}$ are the minimum and maximum lengthening velocities.

## Dynamics of the calcium-force relationship

In module 2 of our muscle-tendon model, the calcium-force relationship was phenomenologically represented due to the lack of experimental data for underlying molecular mechanisms, and the purpose of this study was to investigate the dynamics of calcium sensitivity and cooperativity for force production under physiological conditions. The calcium-force relationship was modulated by dynamically varying the two factors that shape the sigmoid steady-state calcium-activation (CaT/$T_0$-$\tilde{A}_\infty$) relationship (see Fig 1 and Eq (6) for the graphical and mathematical description). One factor (C1) representing calcium sensitivity was the concentration of calcium-binding troponin to reach half of the maximal muscle activation. The other (1/C2) representing cooperativity was for the slope of the calcium-activation relationship at C1. C1 and C2 were mathematically formulated as a sigmoidal function that considers the concentration of calcium-binding troponin relative to the total troponin concentration (i.e., CaT/$T_0$). The equations for the dynamics of C1 and C2 are as follows:

$$\frac{dCX}{dt} = \frac{CX_\infty - CX}{\tau_{CX}} \text{ where } CX_\infty = CXn1 \cdot \left\{ 1 + tanh\left(\frac{\frac{CaT}{T_0} - CXn2}{CXn3}\right) \right\} + CXi \text{ and } \tau_{CX}$$
$$= CXn4$$

where X indicates 1 for C1 and 2 for C2; *CXn1*, *CXn2*, *CXn3* and *CXi* determine the saturation

limit, half maximum, slope and initial value of the $CX_\infty$ curve, respectively; and *CXn4* is the parameter for the time constant of CX dynamics.

## Parameter setting

In this modeling framework, the first module (for calcium dynamics) and the third module (for force production) were constrained by the relevant experimental data. Only the second module (representing the calcium–force relationship) was adjusted for the model to replicate the force data measured under physiological conditions. First, the parameters of module 3 were determined from the data for length- and velocity-tension properties under full excitation. The $K_{SE}$ for the stiffness of serial elastic elements was estimated by dividing the difference in force (i.e., 30.6 N for CAT14 and 9.5 N for CAT12) by the variation in muscle-tendon length (i.e., 1.9 mm for CAT14 and 1.55 mm for CAT12) and then normalized with the peak isometric force ($P_{0.5}$) at $X_{m,0.5}$ (i.e., 100.3 N for CAT14 and 17.5 N for CAT12). $g_1$-$g_3$ for the force-length property were determined by fitting three length-tension datasets (i.e., (0 mm, 62.3 N), (5 mm, 100.3 N), and (10 mm, 115.9 N) for CAT14 and (-0.55 mm, 5.43 N), (3.64 mm, 17.5 N), and (7.9 mm, 30 N) for CAT12) using the function (nlinfit) built in MATLAB software. $a_0$-$d_0$ for the velocity-tension property were determined to capture four velocity-tension datasets (i.e., (-74 mm/s, 57.4 N), (-1 mm/s, 99.3 N), (1 mm/s, 101.3 N), and (71 mm/s, 127.4 N) for CAT14 and (-63.81 mm/s, 1.02 N), (-1.5 mm/s, 17.3 N), (1 mm/s, 17.7 N), and (63.81 mm/s, 36.03 N) for CAT12) using the inverse equations derived in a previous study (Eqs (13)–(16), respectively, in [21]). Second, all parameter values of module 1 except $\tau_1$ and $\tau_2$ for the rate of calcium release from the sarcoplasmic reticulum were adopted from the previous study in [21]. $\tau_1$ and $\tau_2$ of module 1 and *C1i*, *C2i*, *C3-C5* and $\alpha$ of module 2 were simultaneously optimized to fit the shape of the twitch and tetanic responses during isometric contraction at $X_{m,0.5}$ for stimulation frequencies of 10 and 40 Hz for CAT14 and 20 and 40 Hz for CAT12, without considering sag behavior by setting *C1n1* and *C2n1* to 0. The force data obtained at 20 Hz for CAT14 and 10 Hz for CAT12 were not involved because the muscle showed the strongest sag behavior under this stimulation frequency. Third, *C1n1-C1n4* in module 2 were first optimized to replicate the sag behavior observed during the tetanic response at 20 Hz for CAT14 and 10 Hz for CAT12, and then *C2n1-C2n4* were further optimized for the tetanic response at 40 Hz for both CAT14 and CAT12 under isometric conditions at $X_{m,0.5}$. The optimization of parameter values was performed using the optimization tool (i.e., Multiple Run Fitter based on the PRAXIS algorithm) built in NEURON software [39]. Finally, $\varphi_1$-$\varphi_4$ of module 1 for the length-dependent changes in the calcium-force relationship were determined to match the amplitude of the twitch at the physiologically minimal, intermediate and maximal muscle lengths using the polyfit function based on the least squares method built in MATLAB software.

Notably, the model parameters (i.e., *g1-g3* and *a0-d0*) in Module 3 were forced to reproduce the length-tension (L-T) and velocity-tension (V-T) relationship measured under full excitation with 100 Hz frequency stimulation. From the perspective of sliding filament theory [40,41], the values of those parameters reflected the number (N) of force-generating crossbridges formed from spatiotemporal actin-myosin interactions underlying the L-T and V-T relationships. However, for partial excitation with low-frequency stimulation, changes in the fraction of N were reflected by the A-CaT relationship of Module 2. In this study, thus, the number of crossbridges in the force-generating state at a given time corresponds to the product of the maximal tension produced in a given length and velocity (T(L, V)) and A under physiological input conditions.

All parameter values used for the model of the medial gastrocnemius muscle are presented in Table 1.

**Table 1. Parameter values of fast muscle model for cat medial gastrocnemius muscles (CAT14 for model development and CAT12 for model validation).**

| Module | Parameter | CAT14 | CAT12 |
|---|---|---|---|
| | | Value | Value |
| Module 1 | $K1$ [$M^{-1} \cdot ms^{-1}$] | 3000 | 3000 |
| | $K2$ [$ms^{-1}$] | 3 | 3 |
| | $K3$ [$M^{-1} \cdot ms^{-1}$] | 400 | 400 |
| | $K4$ [$ms^{-1}$] | 1 | 1 |
| | $K5i$ [$M^{-1} \cdot ms^{-1}$] | 400000 | 400000 |
| | $K6i$ [$ms^{-1}$] | 150 | 150 |
| | $K$ [$M^{-1}$] | 850 | 850 |
| | $R_{max}$ [$ms^{-1}$] | 10 | 10 |
| | $U_{max}$ [$M \cdot ms^{-1}$] | 2000 | 2000 |
| | $\tau_1$ [ms] | 1 | 12 |
| | $\tau_2$ [ms] | 13 | 19.9 |
| | $\varphi_1$ [$mm^{-1}$] | 0.004 | 0.0716 |
| | $\varphi_2$ | 0.98 | 0.7394 |
| | $\varphi_3$ [$mm^{-1}$] | 0.0002 | 0.0305 |
| | $\varphi_4$ | 0.999 | 0.8889 |
| Module 2 | $C1i$ | 0.154 | 0.159 |
| | $C1n1$ | 0.01 | 0.0032 |
| | $C1n2$ | 0.15 | 0.0094 |
| | $C1n3$ | 0.01 | 0.0083 |
| | $C1n4$ [ms] | 85 | 600 |
| | $C2i$ | 0.11 | 0.109 |
| | $C2n1$ | -0.0315 | -0.0435 |
| | $C2n2$ | 0.27 | 0.315 |
| | $C2n3$ | 0.015 | 0.024 |
| | $C2n4$ [ms] | 70 | 80 |
| | $C3$ [ms] | 54.717 | 54.717 |
| | $C4$ | -18.847 | -18.847 |
| | $C5$ | 3.905 | 3.095 |
| | $\alpha$ | 1.65 | 1.5 |
| Module 3 | $K_{SE}$ [$mm^{-1}$] | 0.16 | 0.35 |
| | $P_{0.5}$ [N] | 100.3 | 17.5 |
| | $g_1$ [mm] | -0.0045 | 0.0003 |
| | $g_2$ [mm] | 0.0981 | 0.1637 |
| | $g_3$ | 0.6211 | 0.4002 |
| | $a_0$ [N] | 0.4 | -34.675 |
| | $b_0$ [$mm \cdot s^{-1}$] | 99.7 | -130.31 |
| | $c_0$ [N] | -57.1 | -57.86 |
| | $d_0$ [$mm \cdot s^{-1}$] | 42.2 | -202.78 |

## Comparison of experimental and simulation data

For the purpose of comparison between different cat MG muscles, all active forces measured experimentally were normalized with $P_{0.5}$ [32]. The error between the experimental and simulation data was quantitatively compared across different levels of current stimulation by calculating the root mean square error normalized with maximal force generation using the

following equation:

$$\text{NRMSE } (\%) = \frac{\sqrt{\frac{1}{N}\sum_{i=1}^{N}\left(F_{m,i} - F_{s,i}\right)^2}}{\left|max(F_m) - min(F_m)\right|} \times 100$$

where the subscripts m and s and N indicate measurement, simulation, and the number of data points (indexed with i) considered for evaluation, respectively.

## Numerical approaches

The model of fast muscle was implemented using the NEURON model description language (NMODL) in the NEURON software environment (version 6.1). The numerical integration of model equations was performed using the integration method (cnexp) built in NEURON with a fixed time step of 0.025 ms. Under this simulation condition, the stability and accuracy of simulations were confirmed while varying parameter values over a wide range. The initial values for $Ca_{SR}$ and CS in the SR and B and T in the SP were set to 2.5 mM, 30 mM, 0.43 mM, and 70 μM, respectively, based on a previous experimental study [33]. The codes for simulations of the muscle-tendon model developed in this study are presented as supplemental information (S1 Data) and also available in the public repository of ModelDB (http://modeldb.yale.edu/267738).

## Results

We constructed a computational muscle-tendon model comprising three submodules to demonstrate whether and how dynamic alterations in the calcium-force relationship of the myofilaments could explain force variations experimentally observed in fast skeletal muscles under physiological conditions (Fig 1). The calcium-force relationship experimentally characterized in skinned muscle fibers was represented as the steady-state relationship of the muscle activation level (i.e., $\tilde{A}_\infty$) to the constant concentration of calcium-binding troponin (i.e., $CaT/T_0$) in the modular model for skeletal muscle (module 2 in Fig 1A). The $CaT/T_0$ for half-maximal activation (i.e., C1 comparable to pCa in the Hill function) and the slope at the $CaT/T_0$ for half-maximal activation (i.e., 1/C2 comparable to Hill coefficient, $n_H$) in the $\tilde{A}_\infty$–$CaT/T_0$ relationship were dynamically varied to infer the dynamic alterations in calcium sensitivity and cooperativity during force production under physiological conditions. The C1 and C2 variations were modulated as a function of $CaT/T_0$ considering the saturation (i.e., *C1n1* and *C2n1*), threshold (i.e., *C1n2* and *C2n2*), and rate (i.e., $\tau_{C1}$ and $\tau_{C2}$) in the modular muscle-tendon model (see Fig 1B for graphical illustration and the Methods for mathematical equations).

### Absence of sag behavior under a static calcium-force relationship

First, it was evaluated whether the model could reproduce the muscle force obtained during isometric contractions at the intermediate length (i.e., $X_{m,0.5} = 5$ mm) between the physiological maximum (i.e., $X_{m,1} = 10$ mm) and minimum (i.e., $X_{m,0} = 0$ mm) over a full physiological range of stimulation frequencies (i.e., 1~40 Hz) with no change in the calcium-force relationship (Fig 2). This condition was implemented by setting the values of *C1n1* and *C2n1* to zero. The parameters (i.e., $\tau_1$ and $\tau_2$ of module 1 and *C1i*, *C2i*, *C3*, *C4*, *C5* and $\alpha$ of module 2) shaping the static calcium-force relationship were optimized to match the experimental data showing no sag phenomenon obtained at 1, 10, and 40 Hz from the medial gastrocnemius of a cat. The model with the static calcium-force relationship could predict the time course of the force profile measured at 1, 10 and 40 as well as 100 Hz. However, the error between the experimental and simulated forces was maximized at a stimulation frequency of 20 Hz, showing the

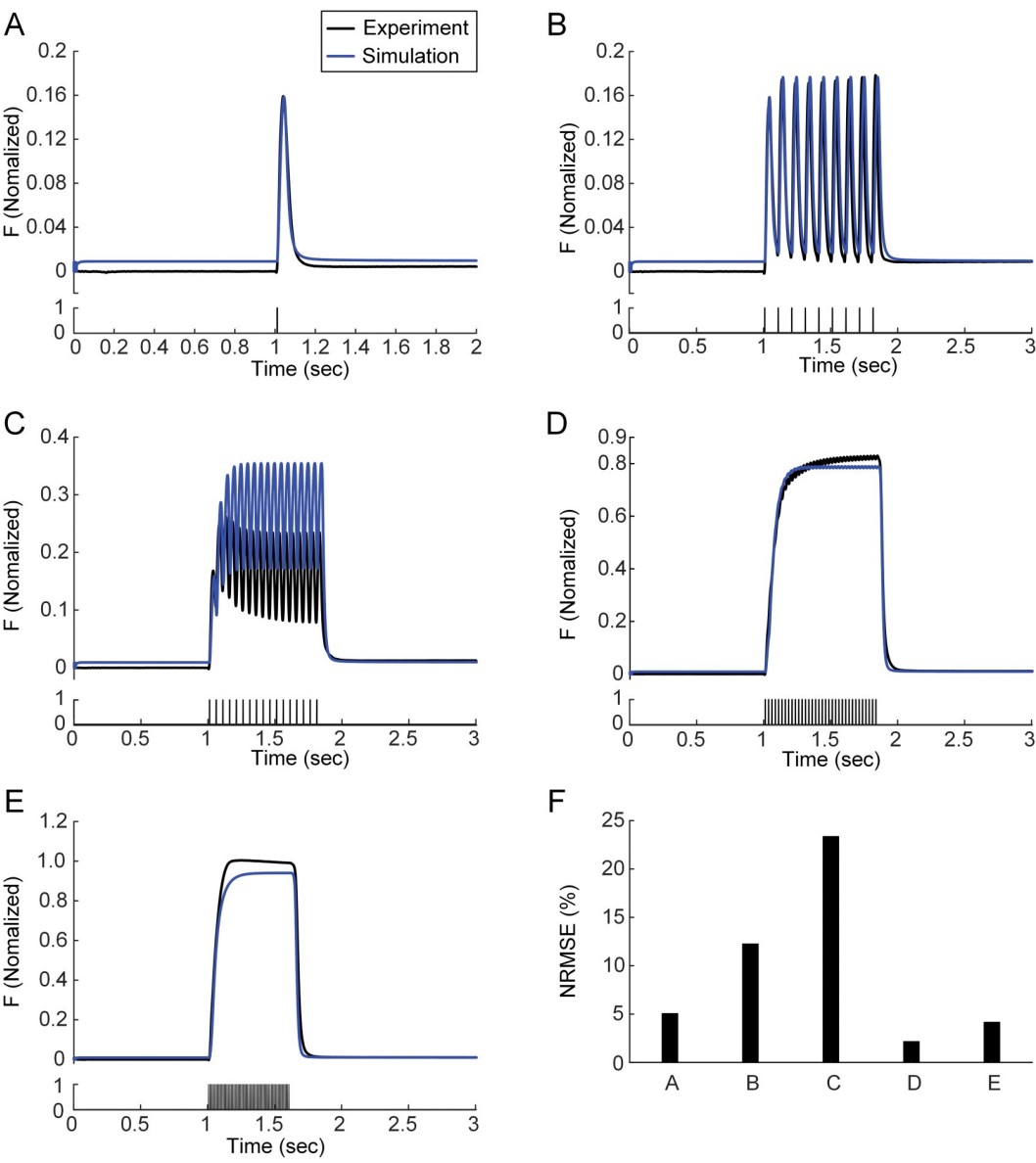

**Fig 2. Tetani of the muscle-tendon model without changes in C1 and C2 in response to various input frequencies under isometric contraction at $X_{m,0.5}$ for CAT14. A.** Twitch (upper) and current stimulation (bottom). **B.** Unfused tetanus (upper) and current stimulation (10 Hz, bottom). **C.** Unfused tetanus (upper) and current stimulation (20 Hz, bottom). **D.** Unfused tetanus (upper) and current stimulation (40 Hz, bottom). **E.** Fused tetanus (upper) and current stimulation (100 Hz, bottom). **F.** Overall error value between simulated and experimental force data for the Panels **A**-**E**. Black and blue lines indicate the data obtained from the experiment and simulation.

absence of the sag phenomenon. This result indicates the insufficiency of the previous model, with no variation in the calcium-force relation, for predicting force production by fast muscle during unfused isometric contractions.

## Depression of calcium sensitivity under low excitation

Then, we assessed whether the error between the experiment and simulation at 20 Hz could be compensated by dynamically varying the calcium-force relationship. The parameters (i.e., *C1n1-*

*C1n4*) underlying the dynamic variation of C1 were optimized for the force data at the stimulation frequency of 20 Hz. The simulation error at 20 Hz could be dramatically reduced by progressively increasing C1 (Fig 3). The variation in 1/C2 alone through parameter optimization (i.e., *C2n1-C2n4*) for the force data at 20 Hz was not sufficient to reproduce the progressive force decline observed during unfused tetanic contractions at 20 Hz (S1 Fig). The decrease in calcium sensitivity, rather than cooperativity, could more accurately predict force production under low muscle excitation levels (i.e., 20 Hz). The dynamic increase in C1 did not affect the force production at stimulation frequencies less than 20 Hz. However, the simulation error was significantly increased under higher levels of stimulation frequency (i.e., > 20 Hz). The dynamic increase in C1 alone was not sufficient to reproduce the time course of force production at 40 Hz.

## Potentiation of calcium cooperativity under high excitation

Finally, we assessed whether the increased error between the experiment and simulation with only variation in C1 for stimulation frequencies > 20 Hz could be compensated by dynamically varying the calcium-force relationship. The parameters (i.e., *C2n1-C2n4*) underlying the dynamic variation of 1/C2 were further optimized for the force data at the stimulation frequency of 40 Hz. The simulation error with only variation in C1 for stimulation frequencies > 20 Hz, including 100 Hz, could be substantially reduced by dynamically increasing the 1/C2 value (Fig 4). The variation in C1 alone through parameter optimization (i.e., *C1n1-C1n4*) for the force data at 40 Hz was not sufficient to reproduce the slow rate of force development following the fast rate of initial force development at 40 Hz (S2 Fig). The gradual enhancement of calcium cooperativity, rather than sensitivity, could lead to better prediction of the slow force development observed at 40 Hz. In addition, the forces at lower stimulation frequencies < 40 Hz could be simulated without the slope variation of the calcium-force relationship by making its threshold a function of the calcium-bound troponin (i.e., CaT) concentration. All these results indicate that calcium sensitivity and cooperativity operationally vary depending upon the state of muscle excitation under physiological conditions.

## Dynamics of the calcium-force relationship at the intermediate length

To visualize how the calcium-force relationship varies depending on the level of excitation, the steady-state calcium-activation (i.e., $CaT/T_0 – \tilde{A}_\infty$) curve was reconstructed at the initial and maximally varied states during isometric contractions at the intermediate muscle-tendon length over the full physiological range of stimulation frequencies (Fig 5A). The shift of the midpoint of the calcium-activation relation to the right was facilitated as the excitation level increased from 20 Hz. However, the slope of the calcium-activation relationship was steepened at higher levels of excitation frequencies from 40 Hz. These findings obtained under steady-state conditions were also confirmed in the transient state for the calcium binding troponin-muscle activation (i.e., $CaT/T_0 – A$) (Fig 5B) and intracellular calcium-muscle force (i.e., Ca–F) (Fig 5C) relationship. The steady-state calcium-activation relationships could be reflected in the transient calcium-force relationships measured during the relaxation of force production (see the insets in Fig 5C). These results imply that the increase in the threshold for crossbridge formation can be counteracted by an enhancement in the cooperativity for crossbridge formation as the level of muscle excitation increases.

## Role of calcium cooperativity in shaping sag behavior

Compared to the experimental data obtained from the cat medial gastrocnemius, we have shown the capability of the muscle-tendon model to produce a typical form of sag behavior

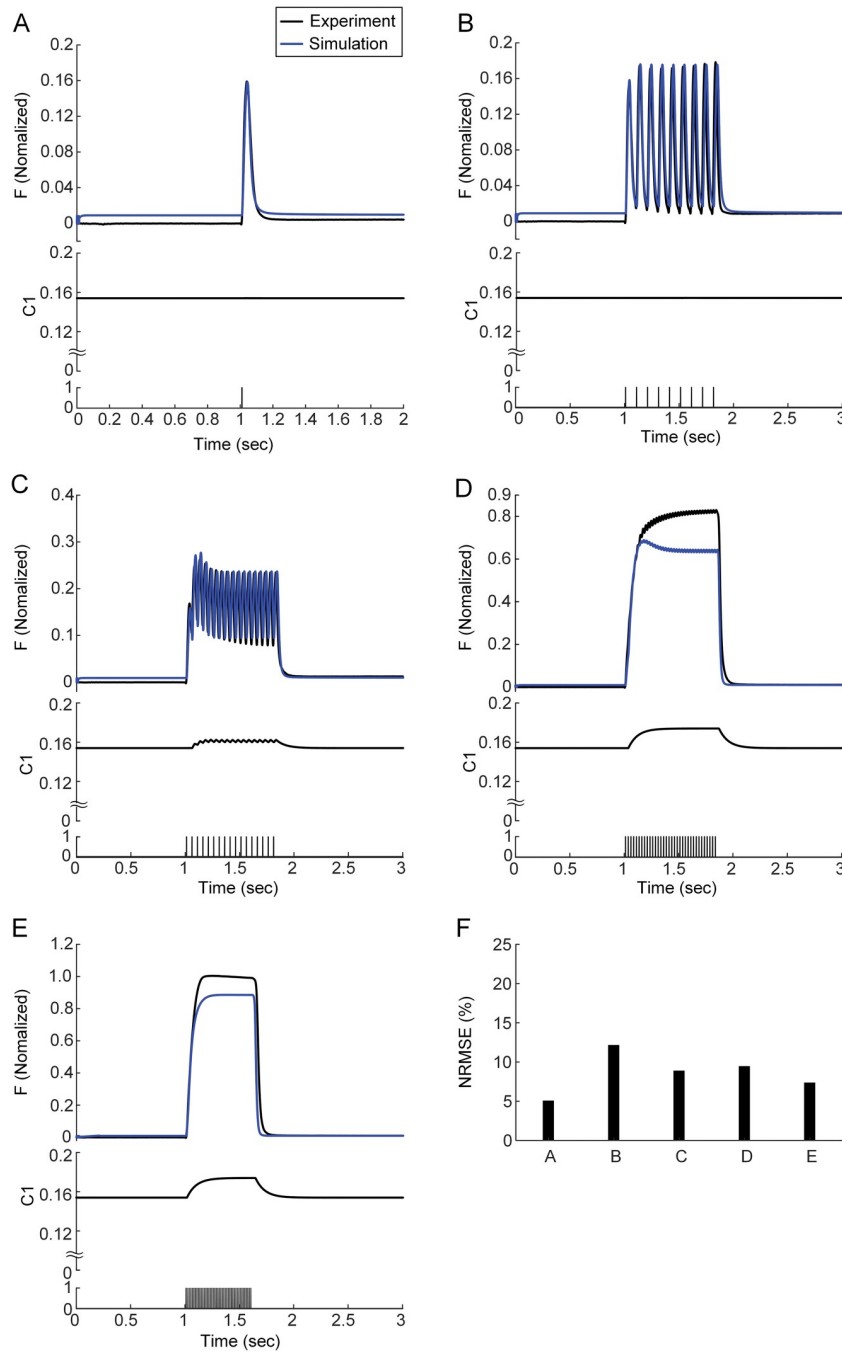

**Fig 3. Tetani of the muscle-tendon model with changes only in C1 in response to various input frequencies under isometric contraction at $X_{m,0.5}$ for CAT14. A.** Twitch (upper), change in C1 (middle) and current stimulation (bottom). **B.** Unfused tetanus (upper), change in C1 (middle) and current stimulation (10 Hz, bottom). **C.** Unfused tetanus (upper), change in C1 (middle) and current stimulation (20 Hz, bottom). **D.** Unfused tetanus (upper), change in C1 (middle) and current stimulation (40 Hz, bottom). **E.** Fused tetanus (upper), change in C1 (middle) and current stimulation (100 Hz, bottom). **F.** Overall error value between simulated and experimental force data for the Panels **A**-**E**. Black and blue lines indicate the data obtained from the experiment and simulation.

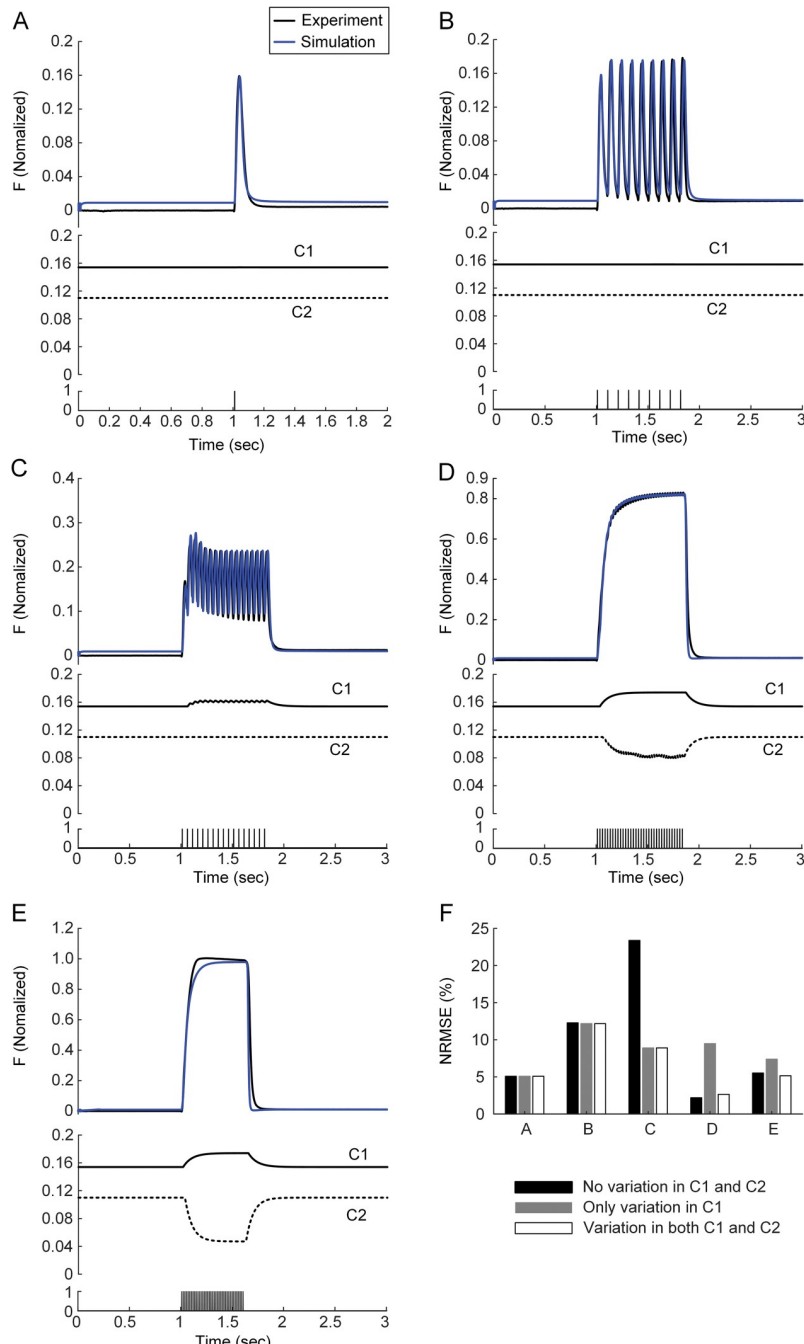

**Fig 4. Tetani of the muscle-tendon model with changes in both C1 and C2 under isometric contraction at $X_{m,0.5}$ for CAT14. A.** Twitch (upper), change in C1 & C2 (middle) and current stimulation (bottom). **B.** Unfused tetanus (upper), change in C1 & C2 (middle) and current stimulation (10 Hz, bottom). **C.** Unfused tetanus (upper), change in C1 & C2 (middle) and current stimulation (20 Hz, bottom). **D.** Unfused tetanus (upper), change in C1 & C2 (middle) and current stimulation (40 Hz, bottom). **E.** Fused tetanus (upper), change in C1 & C2 (middle) and current stimulation (100 Hz, bottom). **F.** Simulation error with no variation in C1 & C2 (black), only variation in C1 (gray) and variation in both C1 & C2 (white) at the stimulation frequency of 1 Hz (A), 10 Hz (B), 20 Hz (C), 40 Hz (D) and 100 Hz (E). Black and blue lines in **A**-**E** indicate the data obtained from the experiment and simulation with variation in both C1 & C2. The comparison of experimental and simulated data for the case of no variation in C1 & C2 and only variation in C1 was presented in Figs 2 and 3.

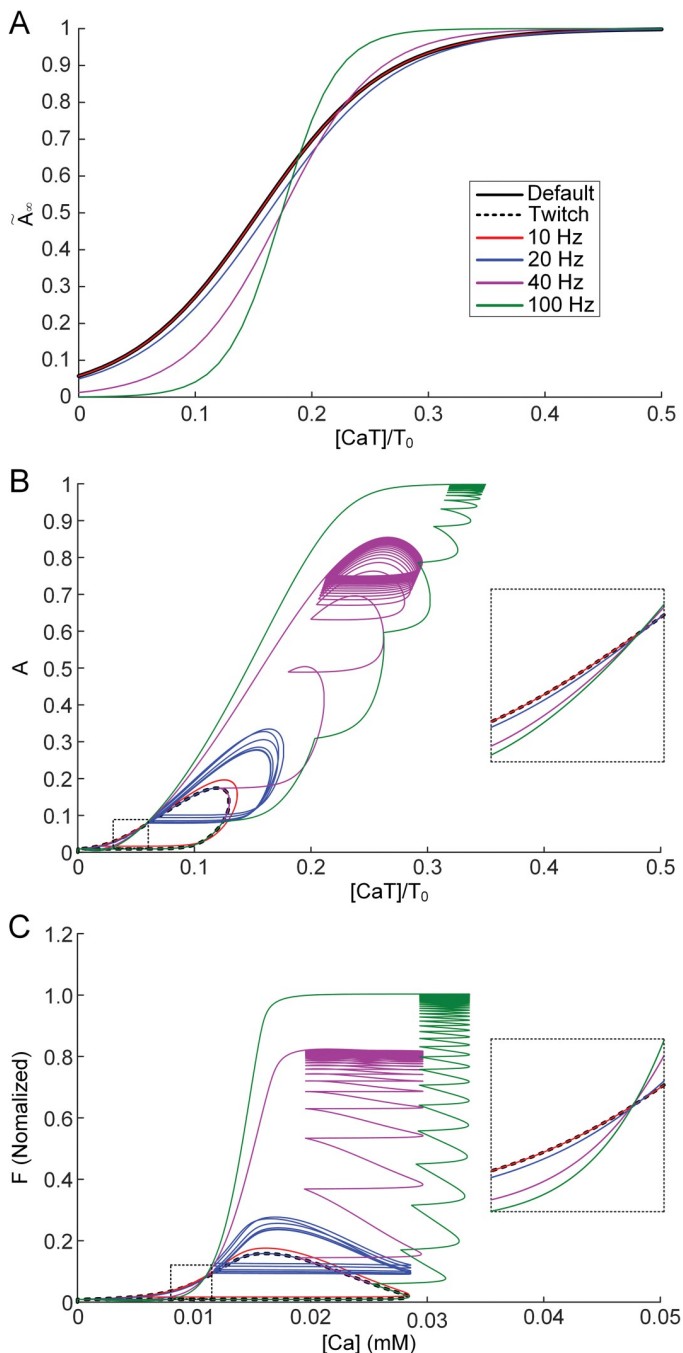

**Fig 5. Variation in the calcium-force relationship during isometric force production at $X_{m,0.5}$ for CAT14. A.**
Steady-state relationship of muscle activation ($\tilde{A}_\infty$) to calcium binding troponin relative to the total troponin
concentration ($CaT/T_0$) in the initial and maximally varied states at various stimulation frequencies. **B.** Transient
relationship of muscle activation (A) to $CaT/T_0$ at various levels of stimulation frequency. **C.** Transient relationship of
muscle force (F) and sarcoplasmic calcium (Ca) at various levels of stimulation frequency. Insets indicate the transient
relationship between calcium and activation (**B**) and calcium and force (**C**) on the relaxation phase of force
production.

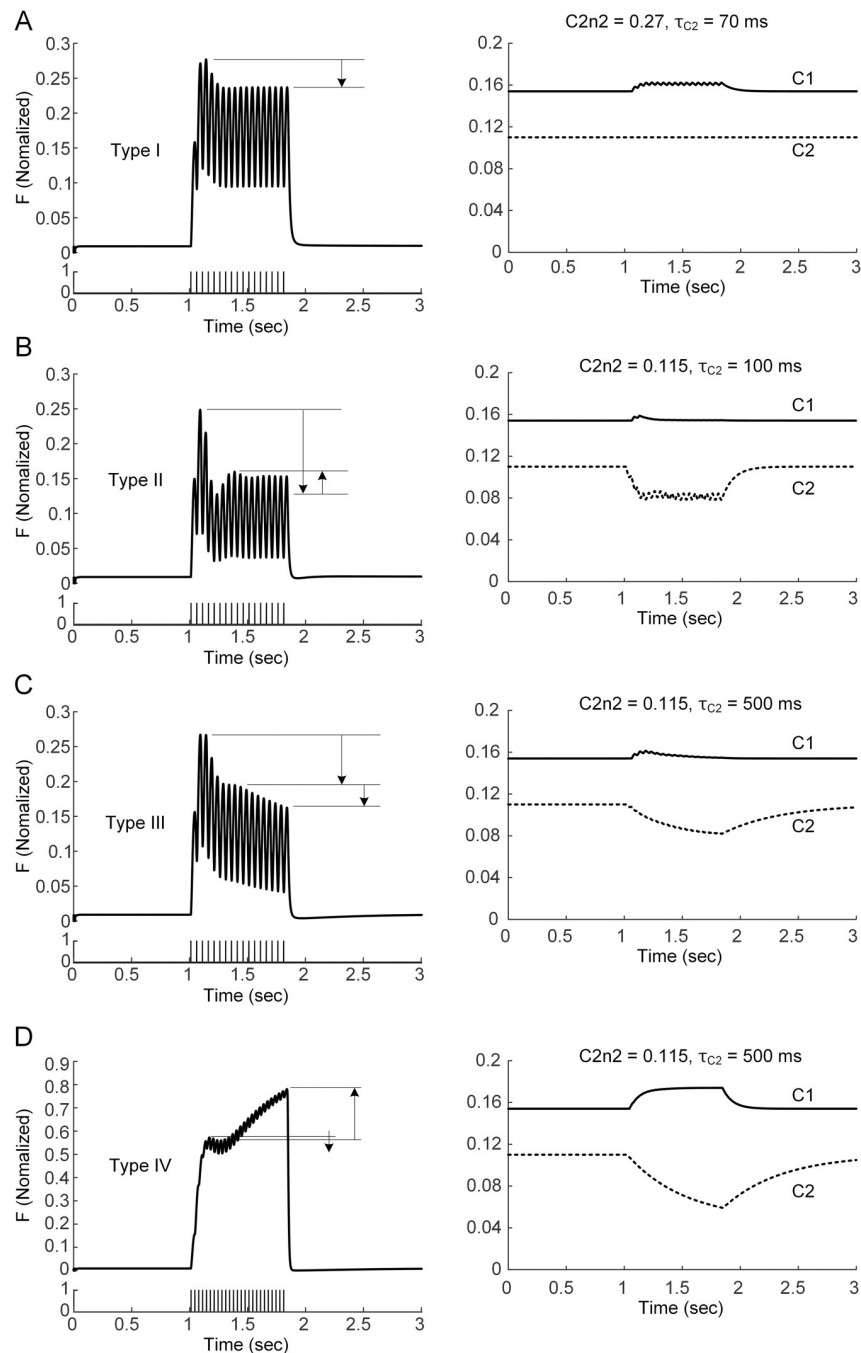

**Fig 6. Representative types of sag behavior at $X_{m,0.5}$ for CAT14. A.** Simple sag form at 20 Hz current stimulation (left) and changes in C1 and C2 (right) with default values of *C2n2* and $\tau_{C2}$. **B-D.** Complex sag forms at 20 Hz (**B** and **C**) and 30 Hz (**D**) stimulation frequencies (left) and changes in C1 and C2 (right) with variations in *C2n2* and $\tau_{C2}$. Arrows indicate the direction of force production after the initial peak force.

showing progressive force decline after the initial peak during the unfused isometric contraction. It was further assessed whether the model could represent more complex forms of sag behavior that have been reported in previous experimental studies [42,43]. Fig 6 shows the four representative types of sag behavior that are identifiable by the model during unfused

isometric contractions at the intermediate length for stimulation frequencies ranging from 20 to 30 Hz. The simple form (referred to as Type I) of sag behavior with progressive force decline after the initial peak could be produced by changing C1 (Fig 6A). However, complex forms of sag behavior could be induced by varying the C2 parameters at the same time. With a lower threshold (less *C2n2*) and slower speed (larger $\tau_{C2}$) (i.e., 100 ms) in the slope variation of the calcium-force relationship, the simple sag behavior transitioned to a complex form (referred to as Type II) of sag behavior, showing multiple force peaks following the initial force peak at 20 Hz (Fig 6B). With the same *C2n2* but much slower $\tau_{C2}$ (i.e., 500 ms) in comparison to the Type II sag, another complex form (referred to as Type III) of sag behavior showing double force declines following the initial force peak was induced at 20 Hz (Fig 6C). With the same values of *C2n2* and $\tau_{C2}$ used for the Type III sag, the last complex form (referred to as Type IV) of force decline followed by force amplification after the initial force peak was determined to be under 30 Hz (Fig 6D). Type I, II and IV sags have been experimentally identified. In contrast, Type III sag theoretically emerged through the analysis of the muscle-tendon model in this study.

We further investigated how *C2n2*, $\tau_{C2}$ and excitation level might interact to shape the sag behavior during submaximal contractions at the intermediate muscle-tendon length. Fig 7 shows the map of sag types over the parameter space for *C2n2* and $\tau_{C2}$ under two different levels (i.e., 20 and 30 Hz) of muscle excitation. Type I sag tended to transition to Type II sag as *C2n2* and $\tau_{C2}$ decreased, while Type I sag tended to transition to Type III sag as *C2n2* decreased and $\tau_{C2}$ increased under a stimulation frequency of 20 Hz. Interestingly, the interaction of *C2n2* and $\tau_{C2}$ on sag type differed under the stimulation frequency of 30 Hz. The sag phenomenon tended to disappear as *C2n2* and $\tau_{C2}$ decreased, while Type I sag tended to transition to Type IV as *C2n2* decreased and $\tau_{C2}$ increased. These results indicate that the type of sag phenomenon might be determined predominantly by a dynamic change in the slope of the calcium-force relationship depending on the level of excitation.

## Calcium-force relationship for the L-T and V-T properties

The dynamic variations in the calcium-force relationship were further estimated for the force production under full excitation (i.e., 100 Hz) while varying the muscle-tendon length at a constant velocity. The muscle-tendon model with variations in the calcium-force relationship could accurately demonstrate the characteristics of force-length and force-velocity properties, as seen in the experiment under full excitation (Fig 8). This result was attributed to the fact that the depression effect of the rightward drift (i.e., C1) of the calcium-force relationship (S3 Fig) was sufficiently compensated by the potentiation effect of the upward drift (i.e., C2) of the slope in the calcium-force relationship under full excitation. As a result, the dynamic variation in the calcium-force relationship did not tend to affect the force production because the activation level was almost maximal under full excitation. This result suggests the importance of the net effect of both changes in the midpoint and the slope of the calcium-force relationship in predicting the length- and velocity-tension properties of fast skeletal muscle under physiological conditions.

## Model predictions for a different cat

We tested whether the results hold for cat MG muscles showing different sag behavior during unfused isometric contractions. The same modeling approach used for CAT14 was applied to the force data independently measured from the different cat MG muscles (i.e., CAT12) with the sag behavior occurring at the lower level of stimulation frequency (i.e., 10 Hz) than CAT14 (see Table 1 for the parameter values of model CAT12). As expected from the model of

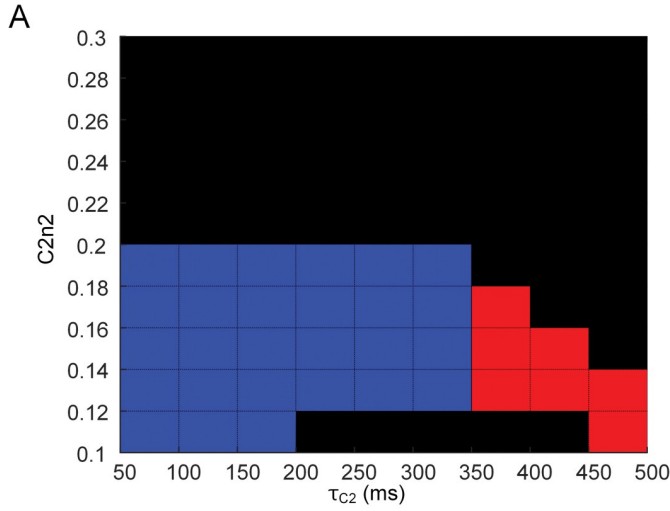

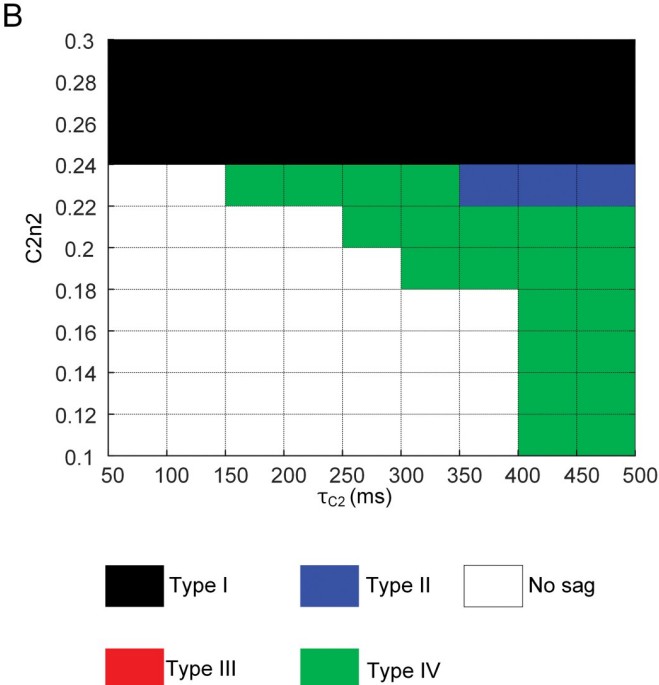

**Fig 7. Type map of sag behavior with changes in threshold (*C2n2*) and time constant ($\tau_{C2}$) for the slope (C2)** **variation in the calcium-activation (CaT/T$_0$–$\tilde{A}_\infty$) relationship at X$_{m,0.5}$ for CAT14. A.** Type map under 20 Hz current stimulation. **B.** Type map under 30 Hz current stimulation.

CAT14, the muscle-tendon model for the new force data could not exhibit the sag behavior without variations in the calcium-force relationship during isometric contractions at the intermediate muscle-tendon length (i.e., X$_{m,0.5}$). The parameters for C1 dynamics (i.e., *C1n1-C1n4*) and C2 dynamics (i.e., *C2n1-C2n4*) were optimized to best represent the new force data at stimulation frequencies of 10 and 40 Hz, respectively. The midpoint (i.e., C1) of the calcium-force relationship was required to drift rightward, whereas the slope (i.e., 1/C2) of the calcium-force relationship needed to drift upward to reproduce the isometric forces over a full range of stimulation frequencies (i.e., 1–100 Hz) at X$_{m,0.5}$, including sag behavior (S4 Fig). The pattern

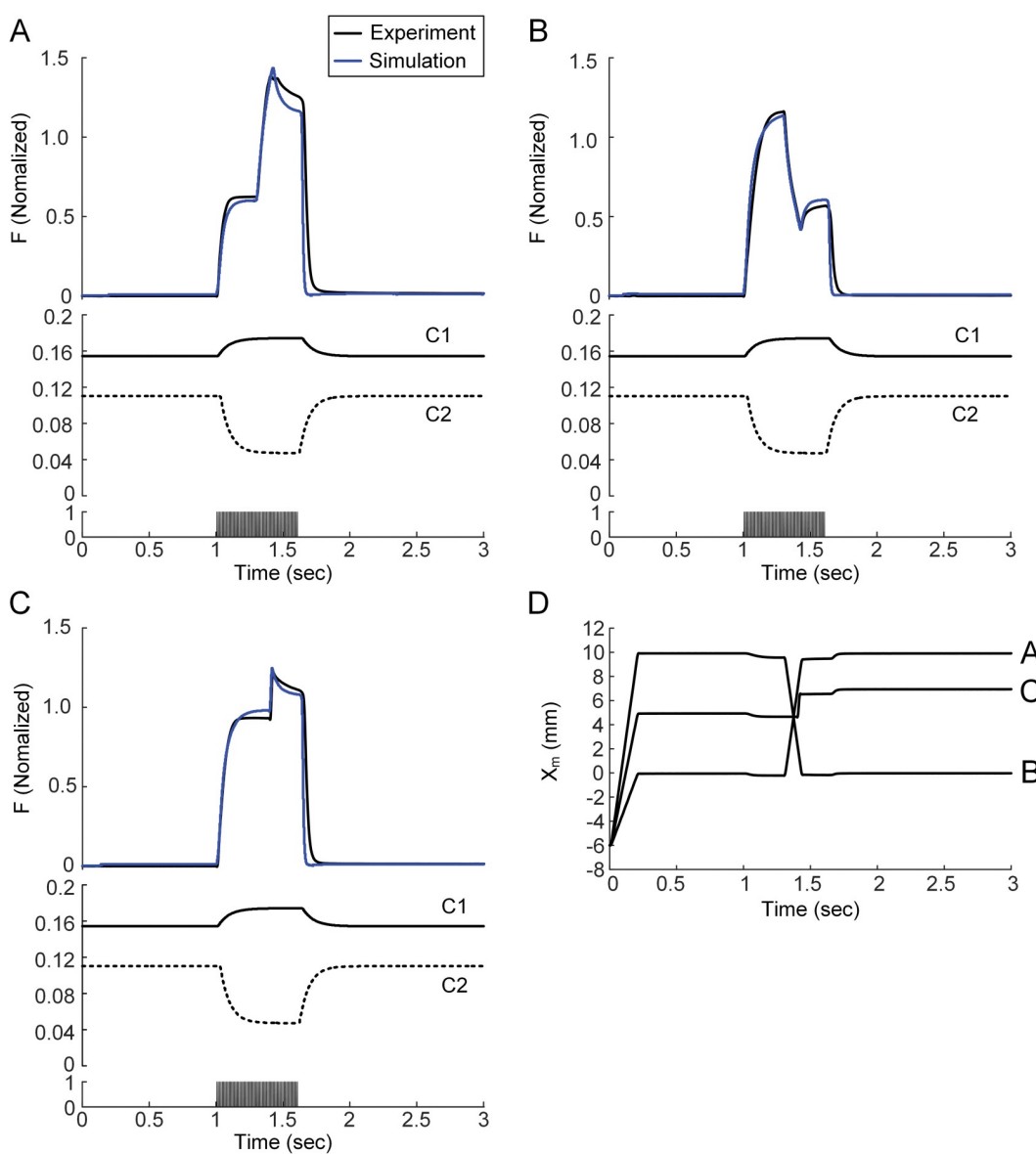

**Fig 8. Length-force and velocity-force properties of the model CAT14 with changes in C1 and C2. A.** Force responses during lengthening (upper), changes in C1 and C2 (middle) and current stimulation (100 Hz, bottom). **B.** Force responses during shortening (upper), changes in C1 and C2 (middle) and current stimulation (100 Hz, bottom). **C.** Force responses during step lengthening (upper), changes in C1 and C2 (middle) and current stimulation (100 Hz, bottom). **D.** Profiles of the muscle-tendon length variation for **A**, **B** and **C**. Black and blue lines indicate the experimental and simulated data.

of dynamic alterations in the calcium-force relationship was similar to that inferred from the dataset for CAT14 (S5 Fig). All types of sag behavior identified by model CAT14 were also found by varying the parameters (i.e., *C2n2* and $\tau_{C2}$) for the slope variation of the calcium-force relationship at different excitation levels (i.e., 20 and 30 Hz) in model CAT12 (S6 Fig). In addition, the model CAT12 reflecting the length dependent twitch response could also replicate the force-length and force-velocity properties under full excitation, showing the importance of the net effect of C1 and C2 variation in the calcium-force relationship (S7 Fig). To this end, the characteristics of dynamic alterations in the calcium-force relationship seem to be similar across fast-type muscles with sag behavior in adult cats.

## Discussion

### Dynamic calcium-force relationship under physiological conditions

The dynamics of the calcium-force relationship during force production differed depending upon the muscle type (slow-twitch versus fast-twitch). In contrast to the case for slow muscles such as the soleus [21], our modular model of the cat medial gastrocnemius muscle predicted that the calcium sensitivity (assessed by C1) and cooperativity (assessed by 1/C2) should dynamically change during force generation over a physiological range of stimulation frequencies (20~40 Hz) and muscle-tendon lengths (0~10 mm) in fast skeletal muscle.

The static calcium-force relationship has been analyzed to infer the calcium dependence of force-producing crossbridge formation under in vitro steady-state conditions [44]. Our study may provide insights into the dynamic aspects regarding the calcium sensitivity and cooperativity of crossbridge formation under physiological conditions. That is, the calcium sensitivity and cooperativity of crossbridge formation might tend to be constant under low levels of muscle excitation (i.e., < 10 Hz) (Fig 2). However, the calcium sensitivity of crossbridge formation might be progressively lowered, resulting in sag behavior under intermediate levels of muscle excitation (i.e., approximately 20 Hz) (Fig 3). The cooperativity of crossbridge formation might be progressively increased, resulting in force enhancement under high levels of muscle excitation (i.e., > 40 Hz) during isometric and isokinetic contractions (Figs 4 and 8). Furthermore, the cooperativity of crossbridge formation might play a crucial role in determining the form of sag behavior (Figs 6 and 7).

All these findings suggest that the calcium sensitivity and cooperativity of crossbridge formation might operationally vary depending on both neural excitation and muscle length in fast muscle under physiological conditions.

### Factors regulating the dynamics of the calcium-force relationship

Contractile force is thought to be determined by the number and force of strongly bound crossbridges and the rate of crossbridge cycling. These crossbridge properties are likely to be regulated dynamically by multiple factors. The rightward drift of the calcium-force relationship might result from a gradual reduction in the number of force-generating crossbridges and the force per force-generating crossbridge under low pH or high inorganic phosphate concentration [45] or in the duration of strong actomyosin binding during crossbridge cycling [19]. The upward drift of the slope of the calcium-force relationship might occur due to a progressive increase in the transition rate from non-force-generating to force-generating crossbridge formation through myosin light chain phosphorylation [46] or in the binding affinity of myosin heads to actin filaments due to mechanosensitive binding proteins [47]. Slow- and fast-twitch muscle fibers have been shown to correspond to Type I and Type II chemically identified by myosin ATPase or MHC I and MHC II by myosin heavy chain isoform [48]. The type-specific compositions of myosin proteins affecting crossbridge cycling have been suggested to underlie slow but sustained contractions of slow muscle fibers and fast but fatiguing contractions of fast muscle fibers [49,50]. These findings from isolated muscle fibers might also explain the differential dynamics of the calcium-force relationship between the fast (Fig 8) and slow (Fig 3 in [21]) muscle models for the length- and velocity-tension relationships under full excitation. The modular modeling framework developed for this study may provide a testbed for further investigation of the relative or combinational contributions of these mechanisms to the calcium-force relationship under physiological conditions.

## Comparison with previous studies

Previous experimental studies have shown a rightward midpoint shift and a decreased slope steepness in the calcium-force relationship as the inorganic phosphate ($P_i$) concentration increases in isolated muscle fibers [18]. As shown in the previous study, the midpoint (i.e., C1) of the calcium-force relationship was required to shift to the right as the excitation level increased in the present study. In contrast to the previous study, however, the slope (i.e., 1/C2) of the calcium-force relationship needed to be steepened as the excitation level increased under physiological conditions (Fig 4). This result suggests that the cooperativity of cross-bridge formation might be enhanced through other factors, such as strain-induced potentiation [51,52], counteracting the inhibitory effects of $P_i$ accumulation at high excitation levels in fast muscle under physiological conditions.

Sag behavior has been investigated in fast muscles in cats [22], rats [42], frogs [53] and humans [54]. A variety of sag forms have been identified from rat triceps surae motor units, including one simple form and two complex forms of the sag phenomenon [42,43]. In this study, our model could reproduce all three forms of the sag phenomenon, suggesting that the simple form (i.e., Type I) tended to be determined by a rightward shift (i.e., C1) of the calcium-force relationship, whereas the complex forms (i.e., Type II & Iv) tended to be shaped by the slope variation (i.e., 1/C2) in the calcium-force relationship (Fig 6). The muscle-tendon model could further identify different types (e.g., Type III) of sag behavior while varying the excitation level and the parameters (i.e., *C2n2* and $\tau_{C2}$) for the threshold and speed in the slope variation of the calcium-force relationship (Fig 7). This result suggests that the interplay between the excitation level and the slope variation in the calcium-force relationship plays a critical role in determining the form of the sag phenomenon.

In intact animal preparations, the sag magnitude has been shown to be greater at shortened lengths than at lengthened lengths [55]. In isolated muscle fiber preparations, the slope of the steady-state calcium-force relationship has been reported to decrease as the muscle length is stretched [7]. Thus, we tested whether the length dependence of sag magnitude could be explained by the slope variation of the calcium-force relationship during isometric contractions at 20 Hz for CAT14 and 10 Hz for CAT12. The model without slope variation in the calcium-force relationship exhibited the increasing sag amplitude with increasing muscle length during isomeric contractions at 20 Hz (Figs 9A1 and 9B1 for CAT14 and S8A1 and S8B1 for CAT12). When the slope of the calcium-force relationship was varied by lowering *C2n2* and decreasing *C2n1* in the shortened state and increasing *C2n1* in the lengthened state from the default value for the intermediate state, as reported from skinned fiber experiments [7], the model muscle displayed the opposite tendency, showing a decreasing sag amplitude with increasing muscle length (Figs 9A2 and 9B2 for CAT14 and S8A2 and S8B2 for CAT12). The length-dependent sag magnitude could be explained by the progressively increased calcium cooperativity at shortened muscle lengths but progressively lowered calcium cooperativity at lengthened muscle lengths under intermediate levels of muscle excitation. This result highlighted the interplay between the slope of the calcium-force relationship and muscle length in shaping the sag behavior under physiological conditions.

The computational model for fast skeletal muscle has been developed phenomenologically, focusing on capturing force-length and force-velocity properties across different levels of muscle excitation [55] or superposition of twitch responses under isometric conditions [56,57]. More recently, it has been proposed that the cooperativity of crossbridge formation should be incorporated into the phenomenological model to reflect the physiological conditions of muscle activation dynamics [58]. In the present study, we have further demonstrated that the dynamic variation in the calcium-force relationship must be considered to reproduce the

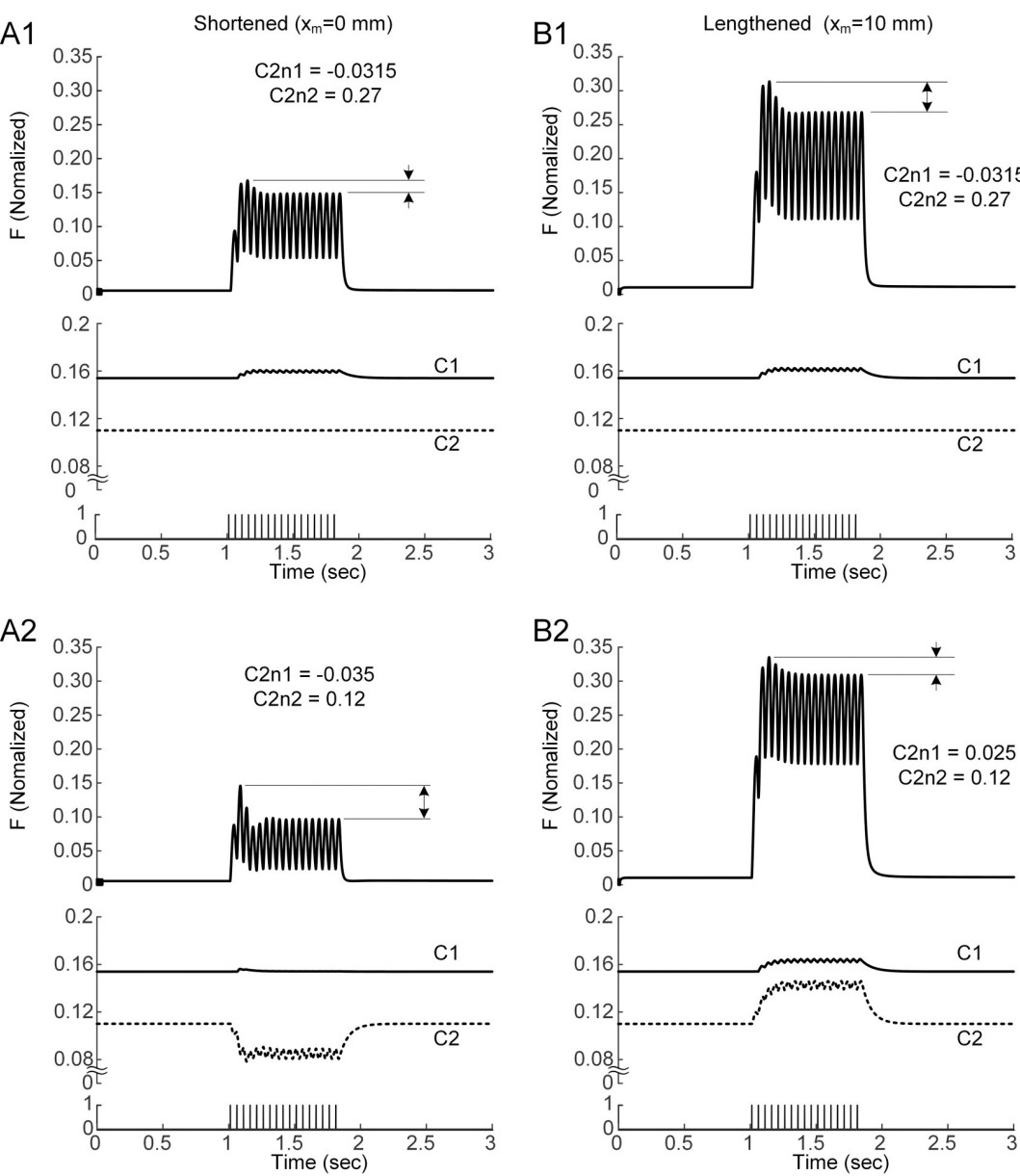

**Fig 9. Length dependence of the sag magnitude in the model of CAT14. A1 & B1.** Unfused tetanus (upper), changes in C1 and C2 (middle) and 20 Hz current stimulation (bottom) with no change in the slope of calcium-force relationship at the physiologically minimal (0 mm) and maximal (10 mm) muscle-tendon length. **A2 & B2.** Unfused tetanus (upper), changes in C1 and C2 (middle) and 20 Hz current stimulation (bottom) with slope variation in the calcium-force relationship at the physiologically minimal and maximal muscle-tendon length. Arrows indicate the degree of force decline after the initial peak force.

complex forms of force production experimentally shown in fast muscle studies under sub-maximal rather than maximal contractions (see Fig 8 for the force-length and force-velocity properties under full excitation). Therefore, the muscle-tendon model developed for this study may provide a physiologically plausible framework for force production over a full physiological range of stimulation frequencies and muscle lengths under isometric and isokinetic conditions.

### Influence of Modules 1 and 3 on sag behavior

In Module 1, we employed a canonical model to transform electrical stimulation into the concentration of calcium binding troponin. This model assumed that calcium ions rapidly occupy the regulatory sites of troponin, and the total troponin concentration is similar in both slow- and fast-twitch muscle fibers [59]. However, type-specific properties, such as the dominant existence of parvalbumin and two regulatory sites of troponin in fast muscle fibers [60], might play a role in determining the sag phenomenon.

We first evaluated the influence of parvalbumin with no variation in C1 and C2 ($C1n1 = C2n1 = 0$). The simulations in Fig 2 were repeated with the same kinetic scheme ($K3 = 4.17*10^7$ $M^{-1}*ms^{-1}$ and $K4 = 0.5$ $ms^{-1}$) and total concentration ($B_0 = 7.5*10^{-4}$ M) as reported in a previous study [60]. The results showed a substantial reduction ($\sim 40\%$ on average) in predicted peak forces, particularly for low stimulation frequencies ($< 40$ Hz), without inducing sag behavior (S9 Fig). Then, we assessed the influence of two regulatory sites of troponin. The reaction equation of $Ca + T = CaT$ was changed to $2Ca + T = Ca_2T$ in Module 1 to explicitly express the interaction between free calcium ions and two troponin regulatory sites (Fig 1A). Accordingly, the system Eqs (3) and (5) were updated as follows,

$$\frac{d[Ca]}{dt} = -(K3 \cdot B_0 + 2 \cdot K5 \cdot T_0 \cdot [Ca]) \cdot [Ca] + (K3 \cdot [Ca] + K4) \cdot [CaB]$$
$$+ \left(2 \cdot K5 \cdot [Ca]^2 + 2 \cdot K6\right) \cdot [Ca_2T] + R - U$$

$$\frac{d[Ca_2T]}{dt} = K5 \cdot T_0 \cdot [Ca]^2 - \left(K5 \cdot [Ca]^2 + K6\right) \cdot [Ca_2T]$$

The total troponin concentration ($T_0 = 1.2*10^{-4}$ M) was set to be the same as that used in a previous study [60]. The twitch response was matched to the force data measured at $X_{m,0.5}$ by increasing the calcium release ($[Ca_{SR}]_{init} = 0.04$ M, $R_{max} = 139$ $ms^{-1}$ and $\tau_2 = 14.5$ ms) from the SR to compensate for the increase in parvalbumin and troponin site concentration. The reaction constants of $K5i$ and $K6i$ were reset to $1.37*10^6$ $M^{-2}ms^{-1}$ and $0.56$ $ms^{-1}$ so that the calcium concentration for half activation ($Ca_{50} = \sqrt{K6i/K5i} = 6.4*10^{-4}$ M) became 70% larger and the calcium-force relationship slope steeper than the canonical case ($Ca_{50} = K6i/K5i = 3.75*10^{-4}$ M). The degree of $Ca_{50}$ increase was determined as the average increase in $Ca_{50}$ in fast muscle fibers compared to slow muscle fibers across rat (100% increase in [61]), monkey (35% increase in [62]), and human (80% increase in [17]) skeletal muscle. As a result, the new model with fast-type-specific properties could not reproduce the sag behavior without variations in C1 and C2 and produced larger forces at high stimulation frequencies ($> 20$ Hz) due to the steeper calcium-force relationship (S10 Fig). However, with $C1i$ (0.162), $C2i$ (0.134), $C2n1$ (-0.0095), and $C2n4$ (200 ms) in Module 2 modestly adjusted, the new fast muscle model could reproduce all isometric and isokinetic force data showing dynamic variations in C1 and C2 similar to those for the canonical case (S11 and S12 Figs).

In addition, calcium binding (K5) and unbinding (K6) to troponin were represented as a function of muscle-tendon length ($X_m$) and muscle activation level ($\tilde{A}$) in Module 1. The isometric forces in Fig 2 were simulated at shortened ($X_{m,0}$) and lengthened ($X_{m,1}$) lengths with and without the length dependence of K5 ($K5i$ for all $X_m$) under no variation in C1 and C2 (S13 Fig). We also simulated the isometric forces at the intermediate length ($X_{m,0.5}$) with and without the activation dependence of K6 ($K6i$ for all $\tilde{A}$) without C1 and C2 variation (S14 Fig). The length dependence of K5 had a modest impact on force predictions only for low stimulation frequencies ($< 40$ Hz) at the shortened state without inducing the sag behavior. However, the lack of activation dependence in K6 caused a considerable reduction ($\sim 85 \pm 0.08\%$,

mean ± SD) in predicted peak forces for all stimulation frequencies without producing the sag behavior.

Furthermore, the present study did not explicitly consider titin-myofilament interactions for predicting active force production during isometric contractions. Instead, extracellular (tendon and aponeurosis) and intracellular (myofilaments and titin proteins) elastic properties were collectively reflected in the serial elastic stiffness ($K_{se}$) in Module 3. In the present study, $K_{se}$ was constant but may vary partially due to changes in titin stiffness upon muscle activation [63]. All simulations in Figs 4 and 8 were repeated over a range of $K_{se}$ values (0.13~0.23 mm$^{-1}$) adopted from a previous study measuring the short-range stiffness as a function of force for cat medial gastrocnemius muscles [64]. The variation in $K_{se}$ over the physiological range had little impact on the predictions of forces during isometric and isokinetic contractions. Notably, titin-myofilament interactions have not affected the length-dependent calcium activation and force-velocity properties [65]. Titin has also been suggested as an active molecular spring underlying the activity-dependent response of the muscle, such as residual force enhancement or suppression after active stretching or shortening [66]. This issue is beyond the scope of the present study focusing on active force production under isometric and isokinetic conditions.

Regarding Module 3, the parallel passive elements were not included in this study. The reason was that the passive force measured without excitation was very low and constant compared to the active force over our investigation's range of muscle length (S15A Fig). In addition, the sag profile, including both active and passive force, was shifted downward when compared with the active force at the same muscle-tendon length (S15B Fig). This result indicates that the dampening and velocity dependency of parallel elements may not significantly impact sag behavior.

All these results reinforce the hypothesis that both calcium sensitivity and cooperativity of crossbridge formation dynamically change to produce the sag behavior observed in fast skeletal muscles under physiological conditions.

## Current limitations

The model including the dynamic variation in the calcium-force relationship could reproduce the force production of a cat medial gastrocnemius muscle over a full physiological range of stimulation frequencies and muscle lengths under isometric and isokinetic conditions. However, this model should be further improved to reflect multiple factors determining the physiological behaviors of fast muscles. First, in the present study, the mechanisms controlling intracellular calcium concentration (Module 1) were assumed not to change during submaximal muscle contractions over short periods of time (i.e., < 1 second). However, this assumption may not hold for maximal muscle contractions over longer periods of time (i.e., > several seconds) since under this condition, the intracellular calcium concentration has been reported to increase and then decrease during progressive force decline or fatigue [24]. Second, the calcium-force relationship was dynamically varied by directly manipulating the parameters of the curve (Module 2), which simulated a progressive increase in the threshold and the cooperativity for actomyosin crossbridge formation. However, the modular modeling approach used for this study could be extended to systematically evaluate potential subcellular mechanisms that might underlie the dynamic variation in the calcium-force relationship during force production in fast muscles [67]. Third, the calcium dynamics in the sarcoplasm were modeled based on the experimental data obtained from rats in the present model. Further experimental investigations are needed to identify the specific differences in sarcoplasmic calcium dynamics between different species and muscle types [68]. Fourth, the modeling approach used for this study represents the average behavior of whole muscle comprising different muscle fibers.

Although cat gastrocnemius muscles are known to be composed predominantly of fast-twitch muscle fibers (i.e., over 75%) [31], this limitation might increase the error between the simulation and experimental data. Future studies should be performed to determine how the heterogeneous organization of whole muscle influences the force output [20]. Fifth, the current model was validated only for constant stimulation frequencies. Neural excitation has been shown to be random rather than constant [69]. Thus, the current model might not reflect the random effect of neural excitation on force production, such as the catch-like property [70]. Finally, the present model is limited to force production under isometric and isokinetic muscle conditions. Future experiments are needed to measure the data on force production during dynamic variations in muscle length, such as locomotor-like movements [29].

## Conclusions

Unlike in slow-twitch muscles, our model suggests that the calcium-force relationship dynamically changes during the force responses of fast-twitch muscles over a full physiological range of stimulation frequencies and muscle lengths. The dynamic variation in the calcium-force relationship operationally varied depending upon the excitation level and muscle length during unfused isometric contractions, whereas it did not tend to affect the length- and velocity-tension properties under full excitation. The slope variation in the calcium-force relationship was crucial to determine the types and length dependence of sag phenomena. These results provide insights into the dynamic properties of crossbridge formation during force production and a computational framework for physiologically realistic modeling of skeletal muscles.

## Supporting information

**S1 Fig. Tetani of the model CAT14 with changes only in C2 during isometric contraction at the $X_{m,0.5}$ (i.e., 5 mm).** Unfused tetanus (upper), change in C2 (middle) and current stimulation (20 Hz, bottom). Black and blue lines indicate the data obtained from the experiment and simulation.
(TIF)

**S2 Fig. Tetani of the model CAT14 with changes only in C1 during isometric contraction at the $X_{m,0.5}$ (i.e., 5 mm).** Unfused tetanus (upper), change in C1 (middle) and current stimulation (40 Hz, bottom). Black and blue lines indicate the data obtained from the experiment and simulation.
(TIF)

**S3 Fig. Impact of the slope steepening of the calcium-force relationship on the force production of the muscle-tendon model used in Fig 8. A-C.** Force production (upper) at current stimulation (100 Hz, bottom) during lengthening, shortening, and step lengthening of the muscle-tendon length, respectively. **D.** Profiles of the muscle-tendon length ($X_m$) variation for **A**, **B**, and **C**. Black, blue and red lines in **A**-**C** indicate the experiment, simulation with and without ($C2n1 = 0$) the slope steepening of the calcium-force relationship, respectively.
(TIF)

**S4 Fig. Tetani of the model CAT12 with changes in both C1 and C2 under isometric contraction at $X_{m,0.5}$. A.** Twitch (upper), change in C1 & C2 (middle) and current stimulation (bottom). **B.** Unfused tetanus (upper), change in C1 & C2 (middle) and current stimulation (10 Hz, bottom). **C.** Unfused tetanus (upper), change in C1 & C2 (middle) and current stimulation (20 Hz, bottom). **D.** Unfused tetanus (upper), change in C1 & C2 (middle) and current stimulation (40 Hz, bottom). **E.** Fused tetanus (upper), change in C1 & C2 (middle) and current stimulation (100 Hz, bottom). **F.** Simulation error with no variation in C1 & C2 (black),

only variation in C1 (gray) and variation in both C1 & C2 (white) at the stimulation frequency of 1 Hz (A), 10 Hz (B), 20 Hz (C), 40 Hz (D) and 100 Hz (E). Black and blue lines in **A**-**E** indicate the data obtained from the experiment and simulation with variation in both C1 & C2. (TIF)

**S5 Fig. Variation in the calcium-force relationship during isometric force production at $X_{m,0.5}$ for CAT12. A**. Steady-state relationship of muscle activation ($\tilde{A}_\infty$) to calcium binding troponin relative to the total troponin concentration ($CaT/T_0$) in the initial and maximally varied states at various stimulation frequencies. **B.** Transient relationship of muscle activation (A) to $CaT/T_0$ at various levels of stimulation frequency. **C.** Transient relationship of muscle force (F) and sarcoplasmic calcium (Ca) at various levels of stimulation frequency. Insets indicate the transient relationship between calcium and activation (**B**) and calcium and force (**C**) on the relaxation phase of force production. (TIF)

**S6 Fig. Representative types of sag at $X_{m,0.5}$ for CAT12. A.** Simple sag form at 20 Hz current stimulation (left) and changes in C1 and C2 (right) with default values of *C2n2* and $\tau_{C2}$. **B-D.** Complex sag forms at 20 Hz (**B** and **C**) and 30 Hz (**D**) stimulation frequencies (left) and changes in C1 and C2 (right) with variations in *C2n2* and $\tau_{C2}$. Arrows indicate the direction of force production after the initial peak force. (TIF)

**S7 Fig. Length-force and velocity-force properties of the model CAT12 with changes in C1 and C2. A.** Force responses during lengthening (upper), changes in C1 and C2 (middle) and current stimulation (100 Hz, bottom). **B.** Force responses during shortening (upper), changes in C1 and C2 (middle) and current stimulation (100 Hz, bottom). **C.** Force responses during step lengthening (upper), changes in C1 and C2 (middle) and current stimulation (100 Hz, bottom). **D.** Profiles of the muscle-tendon length variation for **A**, **B** and **C**. Black and blue lines indicate the experimental and simulated data. (TIF)

**S8 Fig. Length dependence of the sag magnitude in the model of CAT12. A1 & B1.** Unfused tetanus (upper), changes in C1 and C2 (middle) and 20 Hz current stimulation (bottom) with no change in the slope of the calcium-force relationship at the physiologically minimal (-0.55 mm) and maximal (7.9 mm) muscle-tendon length. **A2 & B2.** Unfused tetanus (upper), changes in C1 and C2 (middle) and 20 Hz current stimulation (bottom) with slope variation in the calcium-force relationship at the physiologically minimal and maximal muscle-tendon length. Arrows indicate the degree of force decline after the initial peak force. (TIF)

**S9 Fig. Impact of parvalbumin dominance on the force production of the muscle-tendon model used in [Fig 2](). A-E.** Forces (upper) and current stimulation (bottom) of 1, 10, 20, 40, and 100 Hz, respectively. Blue and red lines indicate the simulation with parvalbumin dominance and that presented in [Fig 2](). **F.** Percent error of peak force produced with the parvalbumin dominance relative to the canonical case in [Fig 2]() for **A**-**E**. (TIF)

**S10 Fig. Impact of two regulatory sites of troponin on the force production of the muscle-tendon model used in [Fig 2](). A**-**E.** Forces (upper) and current stimulation (bottom) of 1, 10, 20, 40, and 100 Hz, respectively. Blue and red lines indicate the simulation with two regulatory sites of troponin and that presented in [Fig 2](). **F.** Percent error of peak force produced with two

regulatory sites of troponin relative to the canonical case in Fig 2 for **A**-**E**.
(TIF)

**S11 Fig. Model predictions with fast-type-specific properties for isometric force production at $X_{m,0.5}$.** **A**-**E.** Forces (upper), change in C1 & C2 (middle), and current stimulation (bottom) of 1, 10, 20, 40, and 100 Hz, respectively. Black and blue lines in **A**-**E** indicate the experimental and simulated force data for CAT14. Gray lines indicate the changes in C1 & C2 of the muscle-tendon model with the canonical model for Module 1 in Fig 4. **F.** Normalized root mean square error (NRMSE) between the experiment and simulation with the fast-type-specific model for Module 1 (black) in the Panels **A**-**E**. NRMSE for the canonical model of Module 1 (white) presented in Fig 4 was overlapped for the purpose of comparison.
(TIF)

**S12 Fig. Model predictions with fast-type-specific properties for isokinetic force production.** **A**-**C.** Force production (upper) at current stimulation (100 Hz, bottom) during lengthening, shortening, and step lengthening of the muscle-tendon length, respectively. Black and blue lines in **A**-**C** indicate the experimental and simulated force data for CAT14. Gray lines indicate the changes in C1 & C2 of the muscle-tendon model with the canonical model for Module 1 in Fig 8. **D.** Profiles of the muscle-tendon length ($X_m$) variation for **A, B,** and **C.**
(TIF)

**S13 Fig. Impact of length dependence of calcium binding to troponin on the force production of the muscle-tendon model used in Fig 2.** **A1**-**A5.** Predicted isometric forces (upper) at the shortened length ($X_m = 0$ mm) under current stimulation (bottom) of 1, 10, 20, 40, and 100 Hz, respectively. **A6.** Normalized root mean square error (NRMSE) between force predictions with and without the length dependence of K5 in **A1**-**A5**. **B1**-**B5.** Predicted isometric forces (upper) at the increased length ($X_m = 10$ mm) under current stimulation (bottom) of 1, 10, 20, 40, and 100 Hz, respectively. **B6.** Normalized root mean square error (NRMSE) between force predictions with and without the length dependence of K5 in **B1**-**B5**. Black and red colors in **A1**-**A5** and **B1**-**B5** indicate the simulation with and without the length dependence in K5.
(TIF)

**S14 Fig. Impact of activation dependence of calcium unbinding to troponin on the force production of the muscle-tendon model used in Fig 2.** **A**-**E.** Predicted forces (upper) and current stimulation (bottom) of 1, 10, 20, 40, and 100 Hz, respectively. **F.** Percent error between peak force with and without the activation dependence of K6 in **A**-**E**. Red, green, and blue colors indicate the full, half, and no activation dependence in K6.
(TIF)

**S15 Fig. Impact of passive parallel elastic elements on the sag behavior. A.** Active and passive force-length relationship with (blue) and without (red) current stimulation (100 Hz) for CAT14. **B.** Normalized force responses (upper) to current stimulation (20 Hz, middle) at the intermediate length ($X_m = 5$ mm, bottom) for CAT14. Blue and red lines indicate the total force produced by the whole muscle and its active force calculated by subtracting the passive force measured without current stimulation from the total force.
(TIF)

**S1 Data. Codes for simulations of the fast skeletal muscle model.**
(ZIP)

## Acknowledgments

The authors truly thank Dr. Thomas Sandercock for collecting data from cat medial gastrocnemius muscles and providing valuable comments on the manuscript.

## Author Contributions

**Conceptualization:** Hojeong Kim, Charles J. Heckman.

**Data curation:** Hojeong Kim.

**Formal analysis:** Hojeong Kim.

**Investigation:** Hojeong Kim.

**Methodology:** Hojeong Kim.

**Resources:** Charles J. Heckman.

**Software:** Hojeong Kim.

**Supervision:** Charles J. Heckman.

**Validation:** Hojeong Kim.

**Visualization:** Hojeong Kim.

**Writing – original draft:** Hojeong Kim.

**Writing – review & editing:** Hojeong Kim, Charles J. Heckman.

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
