## [Decision Letter · Decision Letter 0]

28 Nov 2022

Dear Dr. Kim,

Thank you very much for submitting your manuscript "Modular modeling framework reveals dynamic alterations in calcium sensitivity during force production in skeletal muscle" for consideration at PLOS Computational Biology. As with all papers reviewed by the journal, your manuscript was reviewed by members of the editorial board and by several independent reviewers. The reviewers appreciated the attention to an important topic. Based on the reviews, we are likely to accept this manuscript for publication, providing that you modify the manuscript according to the review recommendations.

Sincerely,

Joanna Jędrzejewska-Szmek, Ph.D.

Academic Editor

PLOS Computational Biology

Jason Haugh

Section Editor

PLOS Computational Biology

Reviewer's Responses to Questions

**Comments to the Authors:**

Reviewer #1: Review is uploaded as an attachment

Reviewer #2: The manuscript entitled “Modular modeling framework reveals dynamic alterations in calcium sensitivity during force production in skeletal muscle” by Kim and Heckman, describes a modification of a previously published modular computational framework, to reproduce the so called “sag behavior” observed in unfused isometric contraction of cats medial gastrocnemius, where fast myosin isoforms are predominat. Because the sag behavior is not observed in slow muscles, the authors propose to modify the previously published modular computational framework designed for cat soleus, where slow myosin isoforms are predominat, introducing a dynamically varying calcium-force relation, specifically introducing a dependence from the actin activation (described by the ratio CaT/T0), in the pCa50 (here called C1) and in the Hill coefficient (here called C2).

With the assumed modification, the model is able to roughly reproduce the force-time behavior at different frequencies optimizing four parameters for each of the two variables, C1 and C2. Based on these evidences the authors deduce several insights about the importance of this mechanism in generating the “sag behavior”, as well as the relative contribution of the two parameters in modulating the sag types.

Major remark:

My major remark is about the impact of the proposed results. The approach proposed in the manuscript is mainly phenomenological, since a number of parameters are modulated to optimize the fitting of a single set of experimental curves (from cat14). The need of modifying almost all the parameters in module 2 (the most relevant ones) to reach a proper fitting for the second set of data (Cat12), doesn’t lead to a validation of the model itself, but more to a sensitivity analysis. Moreover, the model is missing a number of known effects which are influenced by the activation level, force, dynamical behavior etc. Some of them are recognized by the authors as limits, some others are not (for instance the force-driven activation of the the thick filament at sub-maximal thin filament activation, Linari et al Nature 2015). These effects will strongly affect the final behavior of the muscle and greatly limit the possibility of deducing the molecular mechanisms behind the sag behavior. I agree with the authors about the importance of phenomenological approaches to simulate muscle behavior at higher scales in multidimensional models, and that the proposed model can represent a computational framework for physiologically realistic model. However, the manuscript is presented mainly as a way to infer the importance of the dynamic alteration of the calcium sensitivity to explain the sag behavior, see sentences at lines 312 (the necessity), 325 (might decrease), 330 (could not explain), 345 (might be induced), 430 (may be crucial), 471 (was required) and 474 (must be steepened), 500 (we demonstrated), 610 (needed to). In particular I found not appropriate to define Type III sag, which have not been experimentally observed, as a “newly found” type “through the analysis … in this study”.

For the previous considerations, I advance doubts of the impact of the proposed implications deduced by the model analysis. A way to solve this issue could be to rewrite the manuscript reducing the results inferred about the role of the proposed phenomenological approach, and stressing more its possible role in properly predicting the physiological implications of the sag behavior.

Other major remarks:

There are many differences between fast and slow fibers and the modifications of the previously published model, setted for the slow muscle, seem to be inappropriate. Cytosolic buffer (B) in fast skeletal muscle is mainly represented by Parvalbumin, which is virtually absent in slow muscle. The new model simply modify the total amount of “B” but not its kinetics, which I think strongly influence the calcium distribution during the first stimuli. Similarly, Troponin in slow muscle has one binding site, and two in the fast muscle. Also, the values proposed in ref. 37, consider the high and low affinity binding sites, so 0.43 mM, as including the low affinity binding site in the troponin. Authors should refer to www.jgp.org/cgi/doi/10.1085/jgp.201210773 for a better description of the differences between fast and slow models parameters.

The model is based on the assumption that the calcium binding and unbinding to troponin is a function of muscle-tendon length and muscle activation level, which affect the force-time behavior at different frequencies, but a detailed analysis of the role of these parameters, relative the the variations of C1 and C2, is missing. It is important to discriminate the two components in a more quantitative way.

I found the description of the model hard to follow, especially for the need of moving back and forth from the main text to the appendix. I suggest to introduce an extended description of the three modules, with the major equations, in the materials and methods section in the main text.

Minor:

Line 133: L4 and S2 are not defined

Line 155: Xm should be defined here instead of line 181

Line 263: the index “i” is not used

Legends in figures 2, 3, 4, 8 are in panels C, I suggest to move them in the first panel or in every panel.

The definition of the equation for the uptake of calcium (line 17 appendix) seems not familiar to me, a reference or an explanation could help the reader

Reviewer #3: I would like to commend you on the development of this model and the modular approach that you have used. I feel that by creating a module model different aspects and physiological parameters that may lead to SAG can be explored. This type of model also may allow researchers to understand particular elements of SAG and the relative contribution of different proteins, ions, structures and general conditions.

**Have the authors made all data and (if applicable) computational code underlying the findings in their manuscript fully available?**

Reviewer #1: Yes

Reviewer #2: Yes

Reviewer #3: Yes

PLOS authors have the option to publish the peer review history of their article (what does this mean?). If published, this will include your full peer review and any attached files.

Reviewer #1: No

Reviewer #2: No

Reviewer #3: **Yes: **R. John Holash

Figure Files:

Data Requirements:

Reproducibility:

References:

---

## [Decision Letter · Decision Letter 1]

12 May 2023

Dear Dr. Kim,

We are pleased to inform you that your manuscript 'A dynamic calcium-force relationship model for sag behavior in fast skeletal muscle' has been provisionally accepted for publication in PLOS Computational Biology.

Best regards,

Joanna Jędrzejewska-Szmek, Ph.D.

Academic Editor

PLOS Computational Biology

Jason Haugh

Section Editor

PLOS Computational Biology

Reviewer's Responses to Questions

**Comments to the Authors:**

Reviewer #1: The authors have thoroughly and satisfactorily addressed my concerns and questions. In particular, I applaud their willingness to reevaluate their model's performance using alternate variations on the Hill-type muscle model, and their more comprehensive descriptions of their interpretations of their results and their model parameters and their physiological relevance.

The only other feedback that comes to mind is related to the overall context of these findings. Their modeling results suggest that calcium dynamics play a nontrivial modulatory role in fast-muscle force production. Do the authors have any speculations about how role of this modulation in terms of motor control strategies? It could be interesting to include such speculations in the discussion so long as they are explicitly identified as such. However, I don't think this is a critical addition and I certainly don't want to compel ill-conceived speculation especially since this is well outside the scope of this study.

Reviewer #2: The authors have made considerable revisions to the text that help to clarify their findings.

However, in replying to my first major remarks, authors agreed that the modification imposed for the analysis of CAT12 respect CAT14 are too much to consider the process as a "validation". They also correctly modified the subtitle "Validation of the model predictions".

However, the term "validation" is still used in line 130-131 ("CAT14 for model development and CAT12 for model validation". I ask the authors to modify this sentence, that I guess is just linked to the previous version, to make explicit that it cannot be considered a validation.

Except for that, I suggest this paper for acceptance for publication.

Reviewer #3: Thank you for your care and consideration of my comments to the revision of the manuscript I appreciate the time you took to address concerns raised

**Have the authors made all data and (if applicable) computational code underlying the findings in their manuscript fully available?**

Reviewer #1: Yes

Reviewer #2: Yes

Reviewer #3: Yes

PLOS authors have the option to publish the peer review history of their article (what does this mean?). If published, this will include your full peer review and any attached files.

Reviewer #1: No

Reviewer #2: **Yes: **Lorenzo Marcucci

Reviewer #3: **Yes: **R. John Holash

---

## [Editor Report · Acceptance letter]

5 Jun 2023

PCOMPBIOL-D-22-01287R1 

A dynamic calcium-force relationship model for sag behavior in fast skeletal muscle

Dear Dr Kim,

I am pleased to inform you that your manuscript has been formally accepted for publication in PLOS Computational Biology. Your manuscript is now with our production department and you will be notified of the publication date in due course.

With kind regards,

Anita Estes
